# Autoinhibitory sterol sulfates mediate programmed cell death in a bloom-forming marine diatom

Carmela Gallo [1], Giuliana d'Ippolito[1], Genoveffa Nuzzo[1], Angela Sardo[1] & Angelo Fontana [1]

Cell mortality is a key mechanism that shapes phytoplankton blooms and species dynamics in aquatic environments. Here we show that sterol sulfates (StS) are regulatory molecules of a cell death program in *Skeletonema marinoi*, a marine diatom-blooming species in temperate coastal waters. The molecules trigger an oxidative burst and production of nitric oxide in a dose-dependent manner. The intracellular level of StS increases with cell ageing and ultimately leads to a mechanism of apoptosis-like death. Disrupting StS biosynthesis by inhibition of the sulfonation step significantly delays the onset of this fatal process and maintains steady growth in algal cells for several days. The autoinhibitory activity of StS demonstrates the functional significance of small metabolites in diatoms. The StS pathway provides another view on cell regulation during bloom dynamics in marine habitats and opens new opportunities for the biochemical control of mass-cultivation of microalgae.

[1] Bio-Organic Chemistry Unit, CNR- Istituto di Chimica Biomolecolare (ICB) - CNR, Via Campi Flegrei 34, Pozzuoli, NA 80078, Italy. Correspondence and requests for materials should be addressed to G.d. (email: gdippolito@icb.cnr.it) or to A.F. (email: afontana@icb.cnr.it)

Diatoms are ubiquitous photosynthetic components of aquatic ecosystems[1]. Some plankton species develop rapidly into short-term large populations referred to as blooms that are followed by an abrupt decline that results in the export of carbon and silica to the bottom of the ocean[2,3]. Documented cell lysis during demise of algal blooms highlights the importance of natural cell mortality or programmed cell death (PCD) in the dynamics of diatom populations[4–7]. Phytoplankton lysis has been associated with virus infection and bacterial growth[8–10], as well as release of chemical compounds, such as polyunsaturated aldehydes and oxygenated fatty acids, that have been shown to trigger cell death in laboratory cultures[11–14]. Recent evidence suggests that diatoms have sophisticated cell surveillance systems to sense and transduce biotic and abiotic signals at the cellular level[14,15], but the physiological mechanisms controlling the dramatic and abrupt termination of phytoplankton blooms remain very poorly understood[5]. Here we show that a specific class of secondary metabolites, namely sterol sulfates (StS), can act as cellular signals to induce cell death and growth termination of *Skeletonema marinoi*, a marine diatom often dominating spring blooms in temperate coastal waters[16].

## Results

**Effect of natural extracts on *S. marinoi* growth**. Magnitude and longevity of the *S. marinoi* blooms are difficult to model but the species is easy to isolate and maintain in monoclonal cultures that conserve the metabolic characteristics of the original field strains[17–19]. A standard growth curve of diatoms fits a logistic kinetic model with three temporal steps that are by convention identified as exponential (log), stationary, and declining phases. In order to assess the physiological production of intracellular inhibitory compounds, axenic cultures of *S. marinoi* (CCMP2092) were sampled at each of these temporal points and extracted by MeOH (Fig. 1). The resulting extracts were tested at concentrations from 20 to 60 µg mL$^{-1}$ on healthy *S. marinoi* cells

in the exponential growth phase. Cells growth was scored at intervals of 24 and 48 h and normalized for the average number of cells treated with vehicle (control). Extracts corresponding to the declining growth phase (DPEx) impaired the physiological growth of the diatom and induced a drastic reduction in cell number in a dose- and time-dependent manner. After 48 h of incubation, *S. marinoi* cells appeared broken or morphologically altered even at the lowest concentrations of DPEx (Fig. 1d, Supplementary Fig. 13). On the other hand, extracts of cultures at the end of the log phase (ELPEx) and in the stationary phase (SPEx) were weakly or completely inactive, suggesting the formation or release of self-inhibitory factors only in the declining growth phase.

**Identification of sterol sulfates as inhibitory metabolites**. The identification of the active components of DPEx was performed by bioassay-guided fractionation assessing cytotoxic activity on healthy *S. marinoi* cells in 24-multiwell plates. The crude extract was obtained from a 200 L culture of *S. marinoi* (55 g dry weight) harvested in the declining growth phase. After extraction with boiling MeOH[20], the crude extract was desalted and fractionated by a quick run on hydrophobic polystyrene-divinylbenzene resin according to our standard protocol[21]. The fraction eluted with CH$_3$CN/H$_2$O 1:1 was the only one to show activity and was further purified by two sequential steps on reversed-phase HPLC. The whole procedure was repeated twice and led to the isolation of three major compounds that showed in the ESI$^-$ MS/MS spectra a diagnostic fragment at *m/z* 97 due to the loss of a sulfate ion (Supplementary Figs. 1–3) and molecular weights in agreement with the structures of β-sitosterol sulfate (βSITS), dihydrobrassicasterol sulfate (DHBS), and cholesterol sulfate (CHOS) (Fig. 2).

**Chemical characterization of sterol sulfates**. Despite their wide occurrence in nature, it was necessary to confirm the structures of

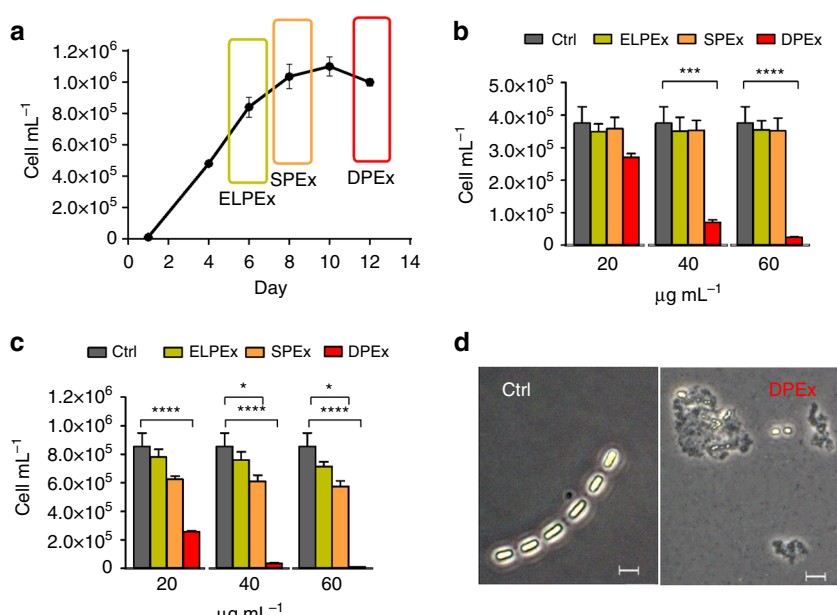

**Fig. 1** Autoinhibitory activity of *S. marinoi* extracts on diatom growth. **a** Growth curve of *S. marinoi*. The rectangular sections indicate the time when cells were harvested and extracted. **b**, **c** Response of healthy *S. marinoi* cells (3 × 10$^5$ cells) after 24 h (**b**) and 48 h (**c**) exposure to growth phase-derived extracts in MeOH (20, 40, and 60 µg mL$^{-1}$). Activity is expressed as cell concentrations (cell mL$^{-1}$) compared to samples treated only with vehicle (Ctrl). Data are means ± s.d. of triplicates of four independent experiments. ELPEx = end of the log phase extract; SPEx = stationary phase extract; DPEx = declining phase extract; Ctrl = control (MeOH); *$P$ < 0.01; ***$P$ < 0.001; ****$P$ < 0.0001 (two-way ANOVA); **d** Formation of cell aggregates in *S. marinoi* cultures after 48 h exposure to 20 µg mL$^{-1}$ of extracts from cells harvested in the declining growth phase (DPEx). Ctrl = untreated cells. Images were taken at ×400 magnification. Scale bars depict 5 µm.

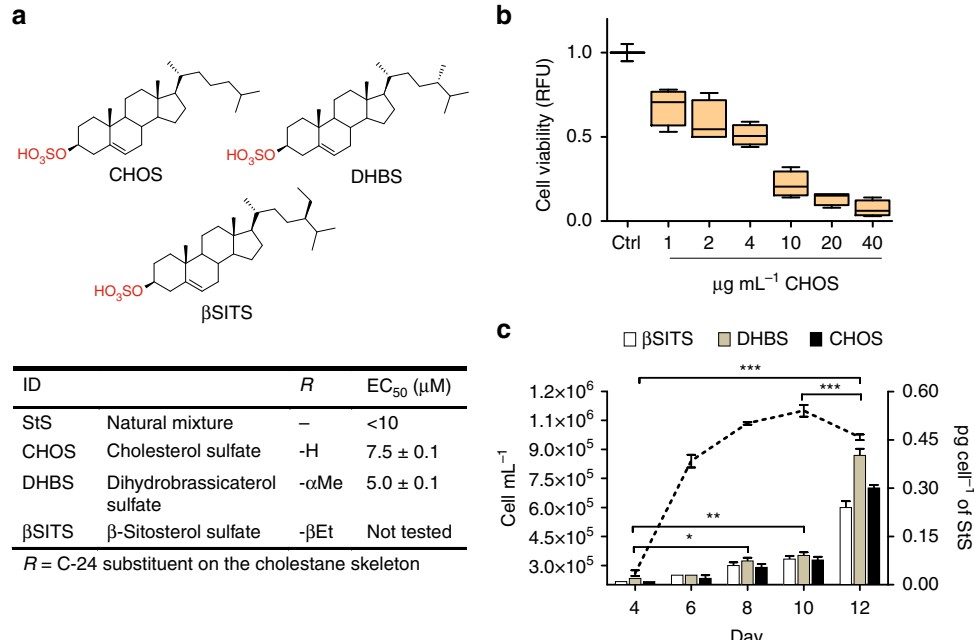

**Fig. 2** Structure and inhibitory activity of pure sterol sulfates from *S. marinoi*. **a** The diatom *S. marinoi* contains three main sterol sulfates with structures corresponding to cholesterol sulfate (CHOS), dihydrobrassicasterol sulfate (DHBS), and β-sitosterol sulfate (βSITS). Natural mixtures of these compounds and the purified products from *S: marinoi* extracts were tested (*n* = 3) at concentrations ranging from 0.5 to 60 μg mL$^{-1}$ on healthy *S. marinoi* cells in log phase in 24-well plates for 48 h. EC$_{50}$ was calculated by linear regression analysis of the logarithm of product concentration vs. mortality rate using standard curve analysis based on a four-parameter logistic in SigmaPlot 11 software. Data are means ± s.d. of triplicates of two independent experiments. **b** Average growth inhibition induced by a synthetic standard of CHOS μg per mL on *S. marinoi* cells at three different cell densities: 1 × 10$^6$, 6 × 10$^5$, and 3 × 10$^5$ cell per mL. Viability at 48 h was determined by the fluorescein diacetate (FDA) assay. Data (means ± s.d.) of Ctrl (DMSO) and CHOS (1, 2, 4, 10, 20, and 40 μg mL$^{-1}$) are expressed as relative fluorescence units (RFU) (*n* = 3). **c** Growth curve and temporal pattern of StS levels (pg per cell) in *S. marinoi* cells. StS were quantified by LCMS, using Cholesterol-25,26,26,26,27,27,27-D$_7$ Sulfate as internal standard. Data are means ± s.d. of triplicates of two independent experiments; *P < 0.05; **P < 0.01, ***P < 0.001 (two-way ANOVA)

the three StS of *S. marinoi* by nuclear magnetic resonance (NMR) and mass spectrometry (MS) analysis (Supplementary Figs. 1–11 Supplementary Table 1). The major product, DHBS, gave a parent ion at *m/z* 479.3207 (calc. 479.3201) for C$_{28}$H$_{47}$O$_4$S$^-$ in the high-resolution electrospray ionization mass analysis in negative mode (HR-ESI$^-$ MS). The $^1$H-NMR spectrum contained 6 methyl signals at δ 0.75 (s, 3H, C-18), 0.83 (d, 3H, *J* = 6.7 Hz, C-26), 0.84 (d, 3H, *J* = 6.7 Hz, C-28), 0.91 (d, 3H, *J* = 6.7 Hz, C-27), 0.98 (d, 3H, *J* = 6.5 Hz, C-21), and 1.07 (s, 3H, C-19), as expected for a 24-methyl sterol with an ergostane C$_{28}$ skeleton. The $^1$H-NMR spectrum also contained the olefinic proton H-6 at δ 5.42 (m, 1H), the aliphatic C-4 protons at δ 2.57/ 2.37, and a signal at δ 4.16 (m, 1H, W1/2 = 9.7 Hz) that was assigned to the axial proton at C-3 bearing the O-sulfate group. The $^{13}$C- and 2D-NMR-experiments confirmed the assignment of the sterol nucleus, as well as the presence of the sulfate group at C-3 (79.8 ppm) and the additional methyl at C-24. The stereochemistry of this substituent was unambiguously established as α in agreement with the analogy with suitable NMR models of dihydrobrassicasterol (24α-methyl cholesterol) and campesterol (24β-methyl cholesterol)[22,23]. CHOS had the molecular formula C$_{27}$H$_{45}$O$_4$S$^-$ as deduced from the HR-ESI$^-$ MS that contained a molecular ion at *m/z* 465.3048 (calc. 465.3044). The $^1$H-NMR spectrum was very similar to that of DHBS except for the presence of only five methyl signals of the classical 5α-cholestane skeleton. The structure was fully confirmed by NMR and chromatographic comparison with an authentic standard.

The third sulfated molecule, βSITS, had the molecular formula C$_{29}$H$_{49}$O$_4$S$^-$ according to the HR-ESI$^-$ MS peak at *m/z* 493.3366 (calc. 493.3357). The $^1$H-NMR spectrum contained six methyl groups including a triplet signal at δ 0.91 (t, 3H, *J* = 7.2 Hz, C-29)

due to the ethyl residue at C-24. The molecule showed the same $^1$H-NMR signals of the other two compounds at C-3 (δ 4.16, half width Wl/2 = 9.5 Hz) and C-4 (2.57, ddd, *J* = 13.0, 4.7, and 2.2 Hz; 2.37, ddd, *J* = 13.0, 12.0, 2.3 Hz in agreement with the presence of the sulfate group at C-3, whereas the remaining spectral data were all consistent with the structure of β-sitosterol.

**Cytotoxic activity of sterol sulfates on *S. marinoi* cells**. The natural mixture of StS showed a cytotoxic concentration (EC$_{50}$) on *S. marinoi* cells below 10 μM, which was in agreement with the activity of the major components DHBS (EC$_{50}$ = 5 μM) and CHOS (EC$_{50}$ = 7.5 μM). We also tested the response of *S. marinoi* cultures at three different cell densities to a synthetic standard of CHOS at concentrations ranging from 1 to 100 μM. The synthetic molecule impaired diatom growth in a dose-dependent manner with severe inhibition of growth at concentrations around 10 μM and greater (Fig. 2b). Notably, the effect was independent of the number of target cells. The experiments indicated that CHOS was per se sufficient to induce cell death, thus supporting the role of StS as intracellular mediators of the process in natural systems. According to these results, target analysis by liquid chromatography–mass spectrometry (LC–MS) revealed a gradual increase in the intracellular concentration of these sulfated metabolites with *S. marinoi* growth progression (Fig. 2c). In the declining phase, the cellular level of StS (0.94 ± 0.08 pg per cell from triplicates of two independent experiments) was five times higher than at the end of the exponential phase and almost 25 times higher than at the beginning of the growth. These results support the growth phase dependent-activity of StS and demonstrate the inhibitory effect in the declining phase when the

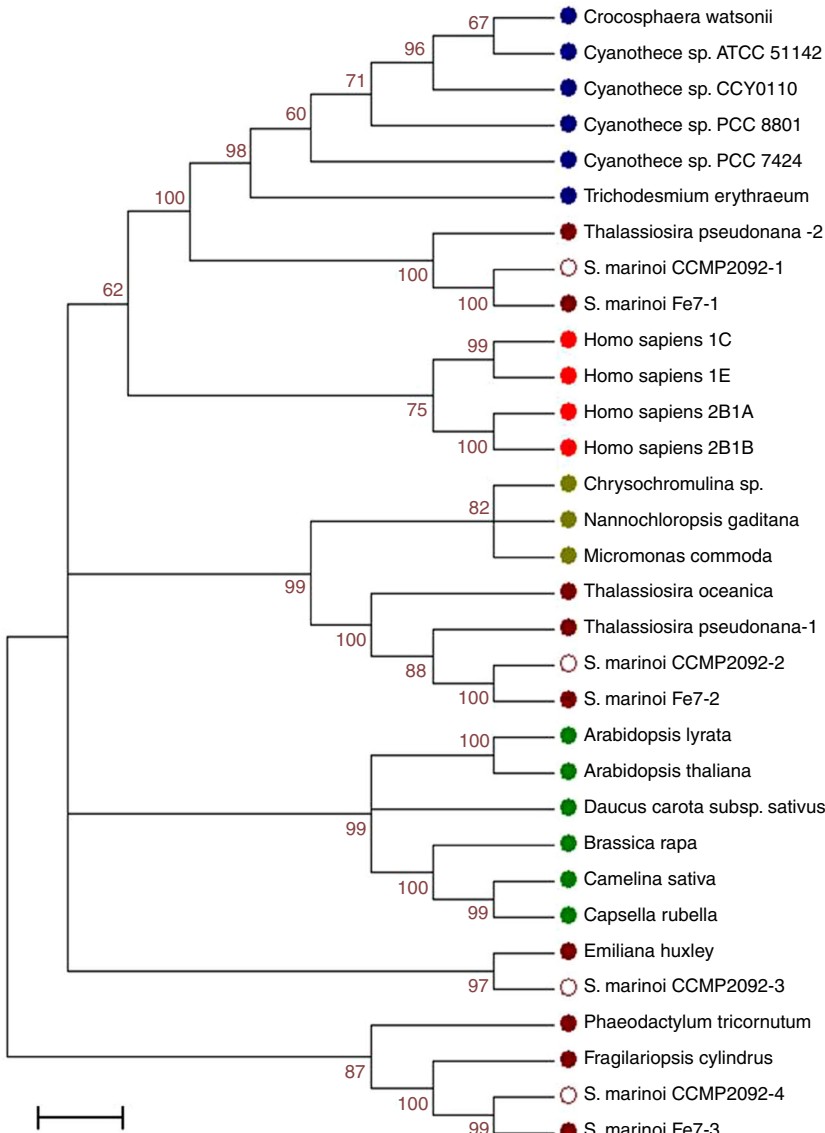

**Fig. 3** Phylogenetic relationship of SULT proteins of diatoms. Phylogenetic analysis of SULTs alignment was inferred by using the Maximum Likelihood method based on the Poisson correction model. The tree with the highest log likelihood (−8017.2374) is shown. The bootstrap consensus tree inferred from 1000 replicates is taken to represent the evolutionary history of the taxa analyzed. Branches corresponding to partitions reproduced in <60% bootstrap replicates are collapsed. The percentage of replicate trees in which the associated taxa clustered together in the bootstrap test is shown next to the branches. Scale bars represent 0.05 substitutions per amino acid position. The four SULT putative sequences (*S. marinoi* CCMP2092-1, 2, 3, 4) were retrieved from the de novo transcriptome assembly of *S. marinoi* (CCMP2092) (Supplementary Table 2). Different colors represent different taxa: blue for cyanobacteria, brown for diatoms, red for human, light green for microalgae, and dark green for plants (Supplementary Table 4)

cellular concentration of these molecules exceeds a threshold level of 0.5–0.6 pg per cell.

**Occurrence of sulfotransferase in marine diatoms.** Sulfonation is an important reaction in the metabolism of endo- and xeno-biotics, including lipids, proteins, and polysaccharides. The reaction is catalyzed by a superfamily of enzymes called sulfo-transferases (SULTs) that have been broadly reported across species[24]. The enzymes are classified in two large groups including membrane-associated members that are mainly responsible for sulfonation of biopolimers and peptides, and cytosolic isoforms that are committed to the sulfoconjugation of small substrates such as steroids and natural products. The addition of a sulfonate group to a compound increases its water solubility and changes its target affinity. Thus, sulfanation

provides an excellent way for the control of hydrophobic signaling molecules, and it is now evident that in many cases sulfoconjugates are not merely derivatives of the free substrates[25,26]. StS have been reported in many organisms but their physiological function has been mostly investigated in humans where the sulfated derivatives of cholesterol and its hydroxylated analogs acquire biological activity that are distinct from the role of the unconjugated steroids[27,28]. Nevertheless, only a few of the mammalian enzymes have been subjected to detailed structural and mechanistic studies[29,30], and there is no report of the systematic study of these proteins across species other than vertebrates. A de novo assembled transcriptome of the *S. marinoi* strain used in this study was mined to identify genes encoding members of the SULT family. The qualitative analysis of the transcripts revealed four nucleotide sequences (TR14735i7, TR5935i1, TR11343i2, and TR8670i1, Supplementary Table 2)

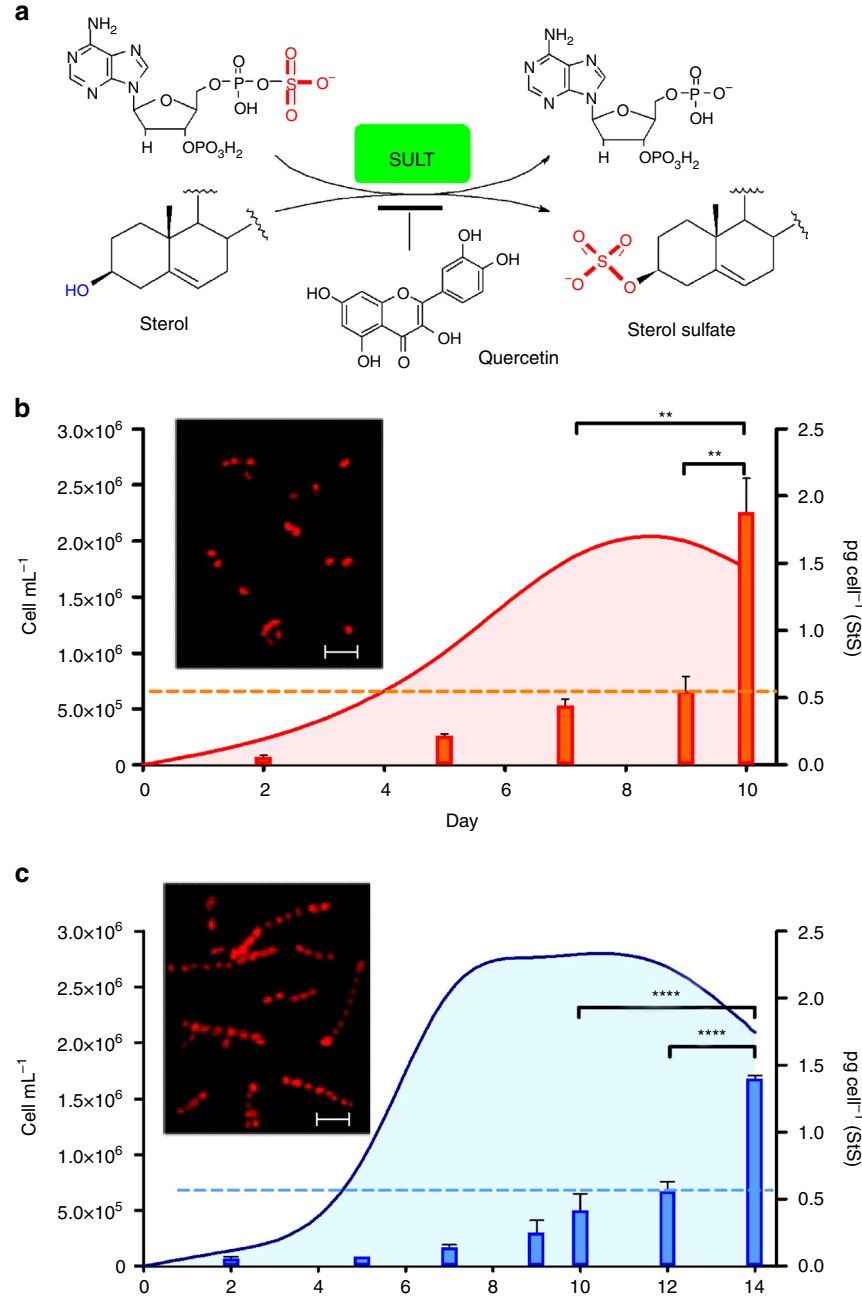

**Fig. 4** Reduction of cellular levels of StS improves growth and duration of *S. marinoi* cultures. **a** Biosynthesis of sterol sulfates by sulfotransferase (SULT) and 3′-phosphoadenosyl-5′-phosphosulfate (PAPS) as donor of the sulfonic group. Inhibition of this reaction by quercetin induces reduction of cellular levels of sterol sulfates. **b**, **c** Response of *S. marinoi* to the SULT inhibitor quercetin. The graphics show cell growth (cell mL$^{-1}$) and cellular concentration (pg per cell) of sterol sulfates in diatom cultures maintained under standard conditions (red) and after addition of 20 µg mL$^{-1}$ quercetin (blue). Cells were counted daily by Burker chamber and chlorophyll *a* fluorescence was measured. Data are means ± s.d. of quintuplicates of three independent experiments; **$P < 0.01$, ****$P < 0.0001$ (Tukey's Multiple Comparison test). Inserts show fluorescence micrographs of control and treated cells in stationary phase (day 9) at ×200 magnification captured by an LP615 emission filter for red chlorophyll autofluorescence. Scale bar depicts 20 µm

coding for four putative proteins (Supplementary Table 3) that contained the conserved *Sulfotransfer-1* domain (PLN02164; Supplementary Table 4). The amino acid sequence analysis also allowed evaluation of the phylogenetic relationship of these proteins within the cytosolic sulfotransferase superfamily in diatoms, cyanobacteria, green algae, plants, and humans (Supplementary Tables 5–6). The maximum likelihood tree analysis shows that the amino acid sequences of *S. marinoi* are most closely related to the SULTs from other diatoms, with the presence of distinct protein families in the lineage (Fig. 3). The four hypothetical proteins of *S. marinoi* share <40% identity but retain 60% or higher similarity with the other members of each homogeneous phylogenetic cluster. We found interesting that *S. marinoi* sequences showed more phylogenetic correlations to the SULTs of cyanobacteria and other microalgae than to SULTs from plants. Notably, the sequence TR14735_i7 encodes for a

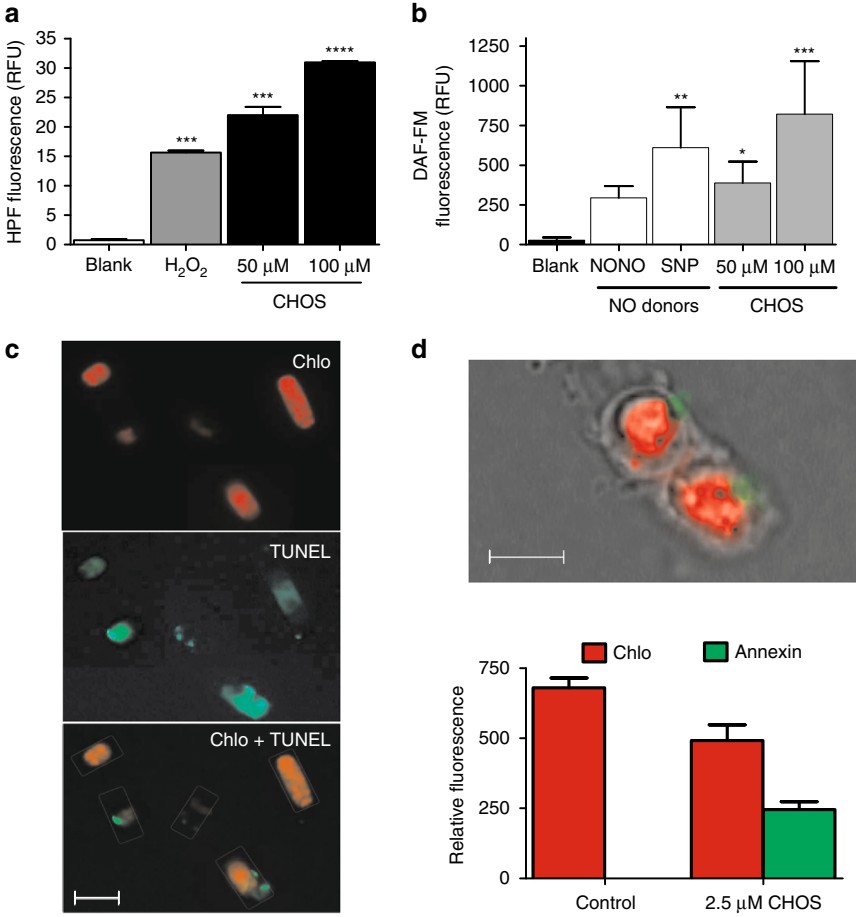

**Fig. 5** Sterol sulfates trigger oxidative burst and programmed cell death (PCD) mechanisms in *S. marinoi*. **a** Production of hROS by exposure of diatom healthy cells to lethal concentrations (50 and 100 μM) of CHOS for 2 h. Activity was determined by the hydroxyphenyl fluorescein (HPF) assay. Data ($n = 6$) are reported as relative fluorescence units (RFU) ± s.d. $H_2O_2$ (200 μM) was used as positive control. Blank = untreated cells; ***$P < 0.001$, ****$P < 0.0001$ (one-way ANOVA of treated cells vs blank); **b** Production of nitric oxide (NO) by exposure of diatom healthy cells to lethal concentrations (50 and 100 μM) of CHOS for 4 h. Activity was determined by 4-amino-5-methylamino-2′,7′-difluorescein diacetate (DAF-FM). Data ($n = 6$) are reported as relative fluorescence units (RFU) ± s.d. The NO donors diethylamine nitric oxide (NONOate) and sodium nitroprusside (SNP) were used as positive control. Blank = untreated cells; *$P < 0.05$, **$P < 0.01$, ***$P < 0.001$ (one-way ANOVA of treated cells vs. blank); **c** DNA fragmentation assessed by TUNEL assay after 24 h application of 2.5 μM CHOS. View under the fluorescence microscope by using 515/565 band filter for the green fluorescence and a LP615 band filters for only red fluorescence emission. Chlorophyll autofluorescence is represented in red (above), fluorescein (TUNEL) in green (middle), merged green and red signal (below). **d** Phosphatidylserine externalization assessed by Annexin V after 24 h application of 2.5 μM CHOS. Composed picture of *S. marinoi* cells under the fluorescence microscope at ×1000 magnification by oil-immersion lens and quantification of the red fluorescence due to chlorophyll and the green fluorescence due to FITC bound to Annexin V. GFP Images were analyzed by ImageJ and normalized to gray value. Data ($n = 9$) are reported as relative fluorescence units (RFU) ± s.d. Control = untreated cells; Scale bars depict 10 μm

putative protein (*S. marinoi*-1) with 30% identity with human cytosolic SULTs (SULT2B1, SULT21A, and SULT1C) that highly prefer cholesterol and other 3β-hydroxysterols as substrates[31].

**Inhibition of sulfotransferase activity in *S. marinoi*.** In humans, SULT isoforms display a strict pattern of structure specificity that is linked to physiological functions. In recent years structural studies have shed light on the catalytic mechanisms of these enzymes and the biochemical determinants of substrate recognition[30,32,33]. However these processes are complex and SULT specificity remains difficult to predict on the basis of protein sequence even after decades of investigation of the individual isoforms. In mammalians, synthesis of CHOS is under control of a SULT subfamily composed of alcohol sulfotransferases that transfer the sulfonate moiety (-SO₃) from the universal donor PAPS (3′-phosphoadenosine 5′-phosphosulfate) to the 3β-hydroxy group of the sterol recipient (Fig. 4a)[24,30]. Specific

research of these reactions has led to the development of a number of inhibitors of human SULTs[29]. No study has been so far carried out in other organisms but potentially these inhibitors can also interfere analogous processes across species. With this idea, we tested seven known inhibitors of human SULTs for their ability to reduce StS levels and affect *S. marinoi* growth (Supplementary Fig. 12). The molecules were added to diatom cultures and the resulting activity was initially monitored by assessment of number and viability of cells in 24-multiwell plates. Of the seven compounds, four (naproxen, spironolacton, salicilic acid, and tetrapropylammonium chloride) were not active whereas clomiphene and damazol were cytotoxic. Only the flavonol quercetin gave a dose-dependent increase in cell growth over nine days. In 10 L cultures, the addition of 20 μM quercetin reduced synthesis of StS and maintained cellular concentration of these molecules below the threshold value of 0.5 pg per cell for a time significantly longer than in control cultures (Fig. 4b, c). This effect resulted in extension of growth duration and increase in cell

concentrations. Furthermore, cultures treated with quercetin did not suffer cell lysis and exhibited longer and a greater number of chains with regularly shaped cells in comparison to controls.

**Apoptotic effect of sterol sulfates on _S. marinoi_ cells.** Diatoms accurately sense environmental changes, including virus infection[8] and nutrient depletion[34–36] via regulatory pathways that eventually control cell fate. Short-chain aldehydes (PUAs) that are produced after cell lysis have been suggested to activate a co-operating mechanism of cell stress and cell death in these organisms[37,38], but to date there is no report on intracellular metabolites that can induce a similar effect in vivo. Thus, in order to investigate the physiological process activated by StS, cells of _S. marinoi_ in exponential growth phase were exposed to lethal concentrations of CHOS for a short time (2 h). Immediately, the diatom responded by increasing the cellular levels of highly reactive oxygen species (hROS) and nitric oxide (NO) (Fig. 5a, b). Both mediators are involved in stress response in plants[39] and have been related to ageing and death in diatom cells[5,14,37]. In _Phaeodactylum tricornutum_, Vardi et al.[40] reported a possible link between NO synthesis and imbalance of redox homeostasis via down-regulation of plastidial MnSOD protein that copes with oxidative stress. Furthermore, both mediators are related to overexpression of death specific proteins (DSP) in the stationary growth phase of _S. costatum_ (ScDSP1)[18,19] and iron starved cultures of _Thalassiosira pseudonana_ (TpDSP1 and TpDSP2)[41]. Recently, these mechanisms have been suggested as having a dual functionality in both acclimation and apoptosis-like processes depending on environmental conditions[37,42]. According to these studies, exposure of healthy _S. marinoi_ cells to a CHOS concentration (2.5 μM) below the natural occurrence in the declining phase caused an evident DNA fragmentation after 12 h of treatment, as detected by TUNEL assay (Fig. 5c, Supplementary Fig. 15). The presence of a multitude of DNA strand breaks is considered to be the gold standard for identification of apoptotic cells, thus the result substantiated a pivotal role of StS in the execution of a cell death program at the concentrations occurring during the declining phase. We also tried to detect early apoptotic events in order to investigate the direct correlation of StS with activation of the cell death mechanism. As reported in Fig. 5d and Supplementary Fig. 14, staining of the cells with Annexin V-FITC

revealed a translocation of phosphatidylserine (PS) from the inner to the outer leaflet within the membrane a few minutes after the addition of CHOS at natural concentrations. However, the specificity of PS externalization is not absolute for apoptosis and loss of the phospholipid asymmetry may be induced by different factors unrelated to the apoptotic response. In particular, although PS exposure apparently occurred without compromising the barrier function of the cell membrane, we could not test rigorously the membrane permeability because the red fluorescence of chlorophyll interfered with the use of cell-impermeable dyes, such a propidium iodide. Under these conditions, the result by Annexin staining cannot be considered conclusive and further experiments are necessary to define the sensitization to cell death induced by StS.

## Discussion

In this study we systematically searched for the occurrence of intracellular small molecules that abruptly terminate diatom blooms at low micromolar levels. The experiments revealed that addition of declining phase extracts or synthetic standard of StS to healthy cultures of _S. marinoi_ triggers clear symptoms of a cell death program. On the contrary, impairment of StS biosynthesis by inhibiting the key sulfonation step protects the diatom and reduces the susceptibility of cell cultures to crash. Identification and quantitation of StS by updated chemical methods underlines a possible threshold effect for these molecules above 0.5–0.6 pg per cell. In fact, intracellular concentrations of StS linearly correlate with culture ageing and reach autoinhibitory levels only in the declining phase, which is in agreement with previous reports that correlate ageing to bloom demise in phytoplankton communities[4,5,14,43]. On the whole, these results imply participation of StS in the regulatory mechanisms of growth and prompt the idea that these small molecules can act as intracellular mediators in signal transduction pathways depending on environmental conditions or physiological changes in diatoms. As shown in Fig. 6, this activity is modulated by accumulation of the inhibitory sterols that induce activation of NO pathways and oxidative burst, as well as sensitize cells to die by apoptotic-like processes. Sterols are major components of cell membranes and their conversion to StS by SULT may involve either de novo biosynthesis or mobilization from the lipid bilayer. SULT expression is under

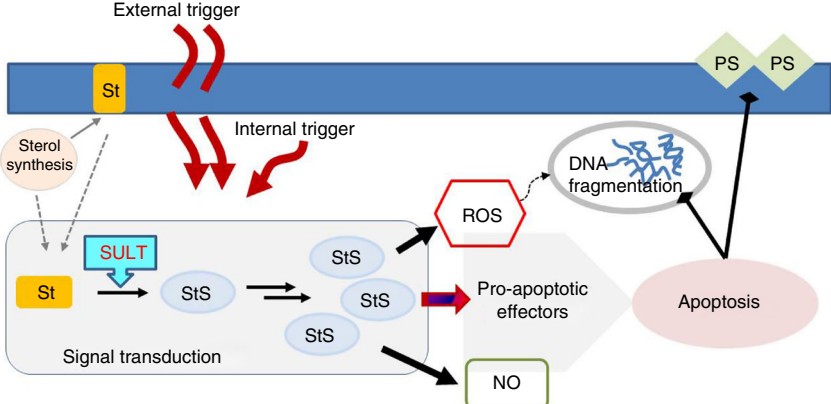

**Fig. 6** Signaling transduction by sterol sulfates in _S. marinoi_. The accumulation of sterol sulfates (StS) has a central role in regulating the diatom cell fate and the associated biochemical responses. Increase of their intracellular levels above a threshold of 0.5–0.6 pg per cell alters redox balance and induces rapid generation of NO. The ultimate effect is the launch of apoptotic events that lead to cell death. The effectors of this last process have not been identified yet but metacaspases are reported in _S. marinoi_ and other diatoms. Synthesis of StS requires sulfonation of the sterol substrates, thus control of the pathway may concern de novo synthesis of sterols, mobilization of sterols from the membrane bilayer or up-regulation of SULT expression. Each of these processes can be triggered by environmental or physiological factors that, depending on the degree and type, determine the intracellular level of StS and, eventually, cell fate. A variety of environmental stressors or chemical molecules, including sterols, can activate SULT expression via specific nuclear receptors according to the literature on SULTs in other organisms

control of numerous members of the nuclear receptor super-family that can be selectively activated by external or endogenous triggers[44]. The overall response to StS displays mechanisms that are reminiscent of cell death in plants where ROS and NO show overlapping and synergistic functions that often induce PCD. However, except for oxylipins and short-chain aldehydes[13,45–49], reports of major roles of small molecules in diatom signaling or regulation are very limited[50,51].

StS have been described only in two diatoms that contain 24-methylenecholesterol sulfate[52,53] and an epidioxy derivative of 24-methyl-$\Delta^{5,7}$-cholestadienol[54] even if they are quite common in other marine organisms. Among microalgae, a $C_{29}$-StS has been described in the haptophyte *Hymenomonas* sp[55]. The molecule is a potent activator of $Ca^{++}$-release and its occurrence in coccolitophores is highly suggestive in the light of the fact that PCD and bloom termination have been reported in other members of this class of microalgae in response to virus-induced infochemicals[8,9]. Starfish[56] and sponges[57], that often feed on phytoplankton, are also a rich and well-characterized source of StS[57] but the origin and physiological function of these molecules is completely unknown. Interestingly, sponges also contain sterols[58] with skeletons that are similar or identical to those described in this study and in previous reports on diatoms[59]. These evident metabolic analogies may underline similarities in the mechanism of cell death in diatoms and these lower metazoans[7]. Sterol sulfonation has never been investigated in marine organisms, but the identification of SULT sequences in all the species of diatoms so far reported in the public NCBI database (*S. costatum*, *P. tricornutum*, *T. pseudonana*, and *T. oceanica*) strongly suggests that this process is commonly present in this lineage of photo-synthetic protists (Fig. 3). In particular, the phylogenetic analysis reveals different SULT families in diatoms and some of these sequences also cluster with those of cyanobacteria and green microalgae. Further studies are necessary to understand the implications of this molecular diversity and identify the enzymes involved in StS biosynthesis in diatoms and their correlation with other marine life forms.

Microalgae are fast growing organisms and have attracted a great deal of attention for the production of biofuels, food commodities, and added-value products[60]. In combination with increased photosynthetic efficiency, the growth rates of microalgae may also contribute to mitigate carbon dioxide accumulation from fossil fuels[61]. However, current technologies have so far failed to meet the expectations of bulk volumes of biomass at low costs and recent interest has focused on the enhancement of mass cultures by manipulation of metabolic pathways. Diatoms have excellent prospects for biotechnological applications as compared to other biomass sources[62]. Dissection and manipulation of StS biosynthesis can offer a new perspective of cellular regulation and present new opportunities to control the growth of these microalgae. Inhibition of sulfotransferase by biochemicals, as shown with quercetin in Fig. 4, or by genetic manipulation can be a valid strategy to enhance biomass yield. It has already been shown that the rational use of selected chemicals can increase productivity of microalgal cultures[63,64]. Apparently sulfonation of sterols does not affect central metabolic processes in diatoms cells, thus use of sulfotransferase inhibitors may find direct and general applications in the development of diatom feedstocks for biotech applications.

## Methods

**General**. NMR spectra were recorded by Bruker AMX 600 spectrometer (Bruker Italia, Milan, Italy) equipped with an inverse TCI Cryoprobe. MS analysis was performed on a micro-QToF mass spectrometer (Water Spa, Milan, Italy) equipped with an elettrospray ionization (ESI) source (negative mode) and coupled with a Waters Alliance HPLC system. High-resolution mass spectra were acquired on a

Q-Exactive Hybrid Quadrupole-Orbitrap mass spectrometer (Thermo Scientific). All HPLC analyses were performed on a JASCO system (PU-2089 Plus-Quaternary gradient pump equipped with a Jasco MD-2018 Plus photodiode array detector) (Jasco, Tokyo, Japan). Fluorescence measurements were recorded by a JASCO Fluorometer FP-8300 equipped with Fluorescence Microplate Reader and an AxioVertA1 (Carl Zeiss, Germany) epifluorescence microscopy with coupled-device camera interfaced with the Axio Vision acquisition/image analysis software. RNA concentration was assayed by a ND-1000 spectrophotometer (NanoDrop) and quality assessed by the Agilent 2100 Bioanalyzer with Agilent RNA 6000 nano kit (Agilent Technologies, Santa Clara, CA, USA). RNA cluster generation and sequencing was carried out by Illumina HiSeq 2500 System (Illumina).

**Microalgae culture and sampling**. *S. marinoi* (CCMP 2092 purchased from Bigelow Laboratories) was grown in f/2 medium at 100 µmol (photons) $m^{-2}s^{-1}$, at $20 \pm 2$ °C, with 14:10 h (light:dark) photoperiod and pH 7 by 50 mM Mops. Replicate cultures were set up in 24-multiwell plates or 10 L-sterile carboys under bubbling with 0.04% $CO_2$ in air (v/v). To accumulate biomass, the diatom was cultivated under fed-batch conditions in 60 L-photobioreactors with 5% (v/v) sterile inoculum under the same conditions. Culture medium pH was automatically adjusted in the range of 7.5–8.5 by 50 mM Mops. Cell density was monitored daily by using a Bürker counting chamber. For quantitative analysis along the growth curve, triplicate samples (1 L) of culture were removed daily without replacement of the medium.

**Purification of microalgal products**. Cells were harvested by centrifugation at $3016 \times g$ for 10 min at 12 °C and were immediately extracted in boiling MeOH[20]. The extracts were then fractionated using CHROMABOND, HR-X resin[21] (Supplementary Methods) followed by reversed-phase high-performance liquid chromatography (RP-HPLC) (Phenomenex, C-18 Luna $10 \times 250$ mm, 100 Å) using a stepwise elution (flow 3 mL $min^{-1}$) from MeOH:$H_2O$ 80:20 (v/v) for 20 min to 100% MeOH in 10 min. Final purification was achieved by a second RP-HPLC step (Phenomenex, C-18 Luna $4.6 \times 250$ mm, 100 Å) using MeOH:$H_2O$ 85:15 (v/v) for 10 min followed by a linear gradient to MeOH:$H_2O$ 90:10 (v/v) in 30 min (1 mL $min^{-1}$). Product elution was monitored by Evaporative Light Scattering Detector (evaporation temperature 45 °C; nebulization temperature 60 °C, $N_2$ flow 1.4 mL $min^{-1}$).

**Chemical analysis of microalgal products**. Purified StSs were analyzed by 1D- and 2D-NMR in 700 µL $CD_3OD$ (600 Hz for $^1H$, 125 MHz for $^{13}C$). LC–MS, MS, and MS/MS spectra were recorded in negative ion mode by using argon as collision gas at a pressure of 22 mbar. The extracts and the purified compounds were dissolved in MeOH to a final concentration of 1 µg $µL^{-1}$ and directly analyzed by LC–MS on a RP column (Phenomenex, C-8 Kromasil $4.6 \times 250$ mm, 100 Å) under the same conditions described for the purification step described above. [25,26,26,26,27,27,27-$D_7$]-CHOS (C/D/N Isotopes) was used as internal standard for quantitation of the products. Calibration curve consisted of blank and standard samples in triplicates at concentrations ranging from 1 to 20 µg $mL^{-1}$. Each concentration point was prepared by successive dilution of a bulk MeOH solution containing 20 µg $mL^{-1}$ of standard. The calibration curve was obtained by plotting peak areas of standard against concentrations. The standard and StS quantifications were carried out by Waters QUANLINK software according to the manufacturer's instructions.

**Chemical data of purified sterol sulfates**. All complete NMR spectroscopic data are reported in the Supplementary Table 1.

CHOS, 5 µg $g^{-1}$ of cell pellet. $^1H$-NMR data ($CD_3OD$, 600 MHz) are reported in the Supplementary Fig. 6; ESI$^-$ MS *m/z* 465.3048 (calc. 465.3044 for $C_{27}H_{45}O_4S^-$, Supplementary Fig. 1).

DHBS, 16.5 µg $g^{-1}$ of cell pellet. $^1H$ and $^{13}C$ NMR data ($CD_3OD$, 600 MHz) are reported in the Supplementary Figs. 4 and 8–11; ESI$^-$ MS *m/z* 479.3207 (calc. 479.3201 for $C_{28}H_{47}O_4S^-$, Supplementary Fig. 2).

βSITS, 5 µg $g^{-1}$ of cell pellet. $^1H$-NMR data ($CD_3OD$, 600 MHz) are reported in the Supplementary Fig. 5; ESI$^-$ MS *m/z* 493.3366 (calc. 493.3357 for $C_{29}H_{49}O_4S^-$, Supplementary Fig. 3).

**Cytotoxic assay**. Organic extracts and fractions were incubated with 1 mL of *S. marinoi* in 24-well plates at concentrations ranging from 0.5 to 60 µg $mL^{-1}$. Fresh cultures and cultures treated with MeOH (vehicle) were used as blank and control. Cells were monitored under the microscope (AxioVertA1; Carl Zeiss; magnifications of ×200 and ×400) and counted in a Bürker chamber. Cytotoxic activity ($EC_{50}$) was calculated using linear regression analysis of the logarithm of product concentration vs. percentage mortality. Experiments and analyses were carried out in triplicate.

**Cell viability**. Diatom cells in exponential growth phase were challenged by CHOS at concentrations of 0.5, 1, 2, 5, 10, 15, 20, 50, and 100 µg $mL^{-1}$ by using fluorescein diacetate [3′,6′-diacetylfluorescein (FDA), Sigma Aldrich][65]. Optimal concentrations and incubation times were assessed in preliminary experiments. A stock

solution of FDA (5 mg mL$^{-1}$ in DMSO) was prepared and stored at 4 °C. Just before use, each aliquot of the stock solution was diluted 40-fold into cold 3.2% NaCl, pH 7.9, kept on ice in the dark, and 25 µL were subsequently injected in 1 mL of the cell culture (FDA final concentration 7.5 µM). Samples were incubated in the dark for 10 min and analyzed by epifluorescence microscopy with magnifications of ×400 and ×1000. FDA fluorescence was monitored at 472/10 nm for excitation and a 512/10 nm for emission. Untreated cultures were used as control. All measurements were carried out in triplicate. Chlorophyll a fluorescence was monitored in quercetin experiments with a microplate reader with a 451/5 nm excitation wavelength and a 679/5 nm emission wavelength.

**hROS detection.** hROS were detected by hydroxyphenyl fluorescein (HPF)[46]. Briefly, S. marinoi cells (about 6×10$^5$ cells per well) were loaded with HPF diluted in sea water (1:8 v/v) and incubated for 30 min in the dark. Samples were centrifuged and washed two times with f/2 medium. The resulting pellets were transferred to 24-well plates with f/2 and treated with CHOS (50, 100, and 200 µM). hROS accumulation was monitored at 1, 3, and 4 h by Fluorescence Microplate Reader at excitation and emission wavelengths of 488/2.5 nm and 515/2.5 nm, respectively. Fresh cultures were used as algal control while cells treated with 200 µM H$_2$O$_2$ were used as positive control. All tests were carried out in triplicate.

**Nitric Oxide detection.** NO was measured by fluorometry using the NO-sensitive dye 4-amino-5-methylamino- 2′,7′- difluorescein diacetate (DAF-FM; Sigma Aldrich) according to Itoh et al.[66]. S. marinoi cells (about 6×10$^5$ cells per well) were incubated with 10 µM DAF-FM in the dark for 30 min. Cells were washed with f/2 medium and kept another 30 min in the dark prior to addition of CHOS (50, 100, and 200 µM) suspended in f/2 medium. Two NO donors, 0.5 mM diethylamine NO (DEANO, Sigma Aldrich), and 0.5 mM sodium nitroprusside, (SNP, Sigma Aldrich), were used as positive controls. Untreated cells were used as blank. Synthesis of NO was monitored at 1, 3, and 4 h with a Fluorescence Microplate Reader at excitation and emission wavelengths of 485/30 nm and 530/30 nm, respectively.

**Phosphatidylserine externalization.** Phosphatidylserine (PS) externalization was detected by staining with Annexin V-FITC (Sigma Aldrich; Italy) according to manufacturer's instructions. S. marinoi cells (1×10$^6$ cell per mL) treated with 2.5 or 5 µM CHOS for 24 h were suspended in 100 µL of Annexin V binding buffer (10 mM HEPES, 140 mM NaCl, and 2.5 mM CaCl$_2$ 2H$_2$O; pH 7.4) and stained with 10 µL of Annexin V-FITC for 20 min at room temperature in the dark. Cells were centrifuged at 3016×g for 10 min and washed twice with cold phosphate-buffered saline (PBS) prior to fixation in 2% formalin–PBS. The images were captured by epifluorescence microscopy (×1000 magnification) equipped with 515/565 BP emission filter for green fluorescence, 525/50 BP emission filter for the green and red fluorescence, LP615 emission filter for the red fluorescence. An unstained control was used for each sample. All assays were carried out in triplicate.

**TUNEL assay.** DNA fragmentation was detected in situ by TUNEL labeling (Supplementary Methods). S. marinoi cells (1×10$^6$ cell per mL) treated with 2.5 µM CHOS for 24 h were fixed in 2% formaldehyde for 20 min[67]. After permeabilization with 3% Triton X-100 (Sigma Aldrich) for 15 min in ice, cells were washed with cold PBS and labeled following the manufacturer's instructions (Roche Diagnostics GmbH). Samples were washed again and suspended in PBS. The images were captured by epifluorescence microscopy as previously described for PS externalization. Fresh algal cultures were used as control while cells treated with 10 µg mL$^{-1}$ DNAse I (Roche, Rotkreuz, Switzerland) were used as positive control. All assays were carried out in triplicate.

**RNA sequencing.** We sequenced, assembled and annotated the transcriptome of three S. marinoi samples from 0.5 L of culture. Total RNA was extracted using the standard RNA extraction method with TRIzol (Invitrogen, Carlsbad, CA, USA). Before use, RNA concentration in each sample was assayed with a ND-1000 spectrophotometer (NanoDrop) and quality was assessed with the Agilent 2100 Bioanalyzer with Agilent RNA 6000 nano kit (Agilent Technologies, Santa Clara, CA, USA). Next generation sequencing experiments, comprising quality control, were performed by Genomix4life S.R.L. (Baronissi, Salerno, Italy). Indexed libraries were prepared from 4 µg/ea purified RNA with TruSeq Stranded mRNA Sample Prep Kit (Illumina) according to the manufacturer's instructions. Libraries were quantified using the Agilent 2100 Bioanalyzer (Agilent Technologies) and pooled such that each index-tagged sample was present in equimolar amounts, with a final concentration of the pooled samples of 2 nM. The pooled samples were subject to cluster generation and sequencing using an Illumina HiSeq 2500 System (Illumina) in a 2 × 100 paired-end format at a final concentration of 8 pmol. The raw sequence files generated (.fastq files) underwent quality control analysis using FastQC (http://www.bioinformatics.babraham.ac.uk/projects/fastqc/).

The highest quality reads from all twelve samples were joined and then used to perform transcriptome assembly with Trinity[68]. The sequences of assembled reads were translated into proteins with Transdecoder (minimum length 100 aa). When multiple translations were possible the priority was set in order to obtain the complete ORF, otherwise the longest sequence was kept when a complete ORF was not detected. To compute abundance estimation, high quality reads were aligned to the Trinity transcripts using bowtie2. Then, RSEM[69] was used to estimate expression values, generating the raw counts matrix and normalized (FPKM) count matrix for each sample (Supplementary Tables 7–8). Differential expression analysis was performed using DESeq[70]. Annotation of transcripts was performed using Trinotate tool of the Trinity pipeline.

**SULT gene analysis.** The de novo transcript sequences of S. marinoi were analyzed by Blast2GO software, performing the functional annotation through homologs search in NCBI-BLAST database, GO term mapping and actual annotation. The analysis was performed in September 2016. Blast2GO output was mined for annotations related to SULT domains and led to identification of four nucleotide sequences (Supplementary Table 2) that were translated to four amino acids sequences by Translate tool (http://web.expasy.org/translate/ -ExPASy-Bioinformatic Resource Tool) (Supplementary Table 3). The four putative SULTs were then aligned by ClustalW2 software and analyzed for conserved domain by NCBI- CD Search Tool. S. marinoi putative SULTs were used as query in NCBI Protein BLAST database to search similar proteins in human and main taxa of photosynthetic organisms (diatoms, plants, green algae, and cyanobacteria). For each taxon we selected only the records with the highest BLAST score having Expecting value higher than 1.00 e$^{-20}$. The selected 32 proteins (Supplementary Table 5) were aligned by ClustalW2 software and were analyzed for the SULT conserved regions previously described in the literature. This alignment was used as input in the phylogenetic analyses conducted in MEGA 7 software by the Maximum Likelihood method. Initial tree for the heuristic search was obtained by applying the Neighbor-Joining method to a matrix of pairwise distances estimated using a JTT model. A discrete Gamma distribution was used to model evolutionary rate differences among sites (2 categories (+G, parameter = 4.1425)). The rate variation model allowed for some sites to be evolutionarily invariable ([+I], 2.2847% sites). The analysis involved 32 amino acid sequences. All positions with <80% site coverage were eliminated. That is, fewer than 20% alignment gaps, missing data, and ambiguous bases were allowed at any position. There were a total of 172 positions in the final dataset.

**Statistical analysis.** The data were analyzed by Graph Pad Prism version 6.0 software. Data were summarized as mean ± SD and between-group differences were tested by ANOVA with correction for multiple comparisons.

**Data availability.** The raw sequencing data from this study have been submitted to the NCBI SRA database (http://www.ncbi.nlm.nih.gov). The RNA sequencing data have been deposited under accession code SRP108217. All other supporting data from this study are available within the article and its Supplementary Information Files, or from the corresponding authors upon request.

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

## Acknowledgements

This study was supported by the project "Integrated Exploitation of Algal Biomass for Production of Energy" (PON01_02740) funded by the National Operational Program for Research and Competitiveness 2007–2013 and by the MIUR-PRIN2015 grant "Top-down and bottom–up approach in the development of new bioactive chemical entities inspired on natural products scaffolds" (Project No. 2015MSCKCE_003). C.G. and A.F. thank Dr Lucio Caso for the technical support and the Doctorate School of Biology (former Applied Biology) of the University of Naples "Federico II". A.F. and

G.dI. acknowledge the RITMARE (Ricerca ITaliana per il MARE) project. We are also immensely grateful to Adrianna Ianora (SZN, Italy) for her comments on the last version of the manuscript.

## Author contributions

C.G. carried out the experiments and planned the technical strategy; G.dI. planned the experimental program; G.N. identified the natural products; A.S. assisted in maintaining the diatom cells and in the preparation of the biological assays; G.d. and A.F. conceived the study and interpreted the experimental results; all the authors assisted in the preparation of the manuscript; A.F. wrote the manuscript.

## Additional information

**Competing interests:** The authors declare no competing financial interests.

