## [Peer Review File · Nature Communications]

Reviewers' comments:

Reviewer #1 (Remarks to the Author):

Diatoms and other phytoplankton are aquatic eukaryotes that generate most of the oxygen in our atmosphere and almost half the primary food chain in oceans. Despite the importance of phytoplankton to life on earth, large gaps exist in our knowledge of the diatom life cycle, which often involves massive blooms that suddenly die off and release large amounts of carbon nutrients into the ocean.

After decades of studying the biochemistry of marine life, Fontana and coworkers have now discovered how sterol sulfates abruptly terminate the diatom blooms at low micromolar levels. The authors systematically searched for a hypothetical small molecule that mediates cell death in the diatom *Skeletonema marinoi*. Experiments involved adding methanol extracts of various cultures to healthy log-phase cultures. Addition of declining phase extracts to healthy cultures triggered symptoms of programmed cell death. Using this assay on chromatographic fractions of the declining phase extracts led to identification of sterol sulfates as substances highly associated with cell death.

This finding is novel. Reports of major roles of small molecules in diatom signaling or regulation are very limited (V. Sonik et al., *Mar. Drugs* 13, 3672 (2015)). No sterol sulfates have been reported in diatoms apart from two vintage papers showing a non-photosynthetic diatom to contain 24-methylenecholesterol sulfate (R. Anderson et al., *BBA* 528, 89 (1978) and *BBA* 573, 557 (1979); this was the only sterol sulfate cited in the 2015 Sonik review). These preliminary results have not been extended during the past 38 years. Descriptive physiology of the rapid die-off of diatom blooms (oxidative bursts, nitric oxide production, DNA fragmentation) is well known, but the triggering of this process has been speculative, including abiotic or viral origins.

The main finding was confirmed and extended: (a) Quercetin, an inhibitor of a human SULT, also inhibited the relevant SULT in *S. marinoi* and thus could be useful in attempts to grow cultures for biofuels or nutrients. (b) Addition of sterol sulfates to *S. marinoi* cultures had the same effects as normal declining phase cultures: fragmented DNA (TUNEL assays), cell aggregates (culture images), and elevated levels of nitric oxide (DAF-FM assay) and highly reactive oxygen species (hydroxyphenyl fluorescein assay).

The extensive experimental work is generally well done and mostly appears adequate to support the stated conclusions. However, I was confused by irregularities in making the phylogenetic tree, a basic error ("40% homology" on page 3, line 97), and the limited detail in how the de novo transcriptome was used to identify the SULT sequences. These methods should be described more accurately in greater detail. Specifically, describe (a) how sequence similarities were calculated, (b) both the full-length and pared sequence alignments used for constructing the phylogenetic tree, (c) the method for constructing the tree, (d) whether sequence identities were nucleotide or amino acid, (e) how the four SULT sequences were identified in *S. marinoi*, and (f) how the transcriptome results were analyzed. Consider calculating 1000 instead of 100 bootstraps and collapsing tree branches with low bootstrap values. Correct errors, such as two different values given for the scale bar.

The manuscript is basically well written, but the authors need to improve minor problems: definite/indefinite article usage, subject-verb plurality, and precise word choice. Getting the correct intended meaning requires author involvement. A few hours with a native English-speaking scientist should suffice to make the ca. 100 needed corrections, including misspellings, typos (e.g. page 7, line 222), and text duplications (page 9, lines 287-292).

Minor points

1. The description of SULTs on page 3, lines 87-89 should be reworded to avoid the impression

that SULTs sulfonate only sterols and always use PAPS. One possibility: "Their synthesis is under control of sulfotransferases (SULT); a subfamily of these enzymes is alcohol sulfotransferases, which transfer the sulfonate moiety (-SO₃) from PAPS (3'-phosphoadenosine 5'-phosphosulfate) to the 3 β -hydroxy group of the sterol recipient."

2. The Figure 3 title is incompatible with panel a; consider moving panel a to the right side, renamed as panel d. Also, maximum likelihood is stated here but neighbor joining is mentioned in the main text (page 10, lines 306-307).

3. The sentence describing MS and NMR characterization of the isolated sterol sulfates states that "their structures were fully elucidated by 2D-NMR analysis (Supplementary Fig. S1-S4)". However, these figures show only 1D proton NMR spectra, with no numerical chemical shift information. For the benefit of researchers pursuing the growth of diatoms for biofuels and needing to chemically synthesize sterol sulfates, a typical list of NMR chemical shifts and spectral figures would be useful. Such supporting information is standard in synthetic and natural products journals, such as *J Org Chem* and *J Nat Prod*. Incidentally, the NMR figures indicate high chemical purity and high spectral quality.

Reviewer #2 (Remarks to the Author):

General:

The manuscript describes isolation of three sterol sulfate molecules from a single marine diatom species and describes their effect on cell viability in culture. Potential genes involved in their synthesis are identified and speculation is made about their roles in broader diatom ecology. The work is novel, interesting, appears to be well done, and, once placed in proper context should prove valuable and stimulatory. There are, however, a number of significant issues.

1) The terminology and framework surrounding the processes of cell death are confused and confusing. This needs to be cleared up throughout the manuscript. For example, in the abstract, "Natural mortality" is described as a "physiological mechanism", yet the vast majority of phytoplankton ecologists would include processes like grazing and sedimentation under 'natural mortality'. Terms like "cell lysis", "programmed cell death" and "apoptosis-like death" crop up and are used, almost interchangeably in places. What needs to be made clear is that: a) cell death in algae can be triggered by intrinsic as well as extrinsic factors, and b) the cell can actively participate in its own death. Whatever terminology the authors select, they should use it consistently. For one example of a consistent framework, see: Franklin, D., C. Brussaard, and J. Berges. 2006. What is the role and nature of programmed cell death in phytoplankton ecology? *European Journal of Phycology* 41: 1-14.

2) The background to the work is patchy. a) For example, though the work of Vardi et al. (2009) is cited, the more recently developed viral-glycosphingolipid-cell death story isn't commented on (Vardi et al. 2012; Host-virus dynamics and subcellular controls of cell fate in a natural coccolithophore population. *Proc Natl Acad Sci USA* 109: 19327-19332). These authors demonstrated the induction of reactive oxygen species, caspase-specific activity, metacaspase expression, and programmed cell death in response to the accumulation of virus-derived glycosphingolipids upon infection of natural *E. huxleyi* populations. The lipid-cell death connections seem important. b) We are told nothing about sterols in diatoms, despite the rich literature (especially from John Volkman's group, e.g. Barrett et al. 1995. Sterols of 14 species of marine diatoms. *J. Phycol.* 31: 360-369)...what are these compounds doing and why would they be appropriate/useful signaling compounds? c) It isn't mentioned that marine sponges (many of which graze on phytoplankton) are the richest and best-characterized sources of sterol sulphates, and the contention that steroid sulphate function "has been accurately investigated only in humans" (line 86-87) isn't quite accurate: Kobayashi et al. 1989. Hymenosulphate, a novel sterol sulphate with Ca-releasing activity from the cultured marine haptophyte *Hymenomonas* sp. *J. Chem. Soc. Perkin Trans. 1*: 101-103.

3) Culture growth phases and growth calculations. In fact, the "log" phase cultures are demonstrably not in "log" phase. Consider Figure 1. a, plotted on normal-normal axes. From day 2 to 3, cell numbers go from about 2×10^5 to 4×10^5 , an approximate doubling. If the cultures were in log phase, we would then expect another doubling in the next day...to 8×10^5 by day 4. In fact, this isn't achieved until day 8. These cultures are simply not growing logarithmically by any definition. This would actually be clear if plotted on log-normal axes, a convention that is often recommended for culture data. The authors need to find a new description for the phase of growth of these cultures. In terms of growth rates expressed as a percentage of controls, it's not immediately apparent how negative numbers can be achieved (Figure 1, b and c). How were growth rates calculated? Are these in fact cell loss rates? What units are being used before they are scaled to the controls, e.g. d^{-1} using a natural or a base 10 log? This is entirely unclear.

4) The phosphatidylserine externalization experiments lack a proper control. Annexin V is a small molecule and will diffuse across 'leaky' cell membranes. Cell membranes become compromised as cells die and thus Annexin V become non-specific because cells stain *from the inside* without any externalization. The normal positive control is to counterstain with propidium iodide (PI). Cells staining with both PI and Annexin V are false positive, while those staining only with Annexin V can be considered to show phosphatidylserine externalization. There is no evidence that such a control was done, so the results cannot be interpreted and should not be presented.

Specifics:

5) Title: "marine diatoms" is inappropriate: the authors have demonstrated an effect in a single species. They cannot generalize like this.

6) Sterol sulphates are described using trivial names; descriptions should also include proper IUPAC nomenclature...this is critical for comparisons with the literature.

7) Figure 2. b. The units of growth rate are missing from the Y axis.

8) Reference formatting needs attention. Many species names are not italicized and the titles of references are not consistently capitalized.

9) Throughout the manuscript there are grammatical issues with missing adjective articles ("a", "the"); though annoying, these don't generally affect interpretation.

Reviewer #3 (Remarks to the Author):

I believe this manuscript provides the first evidence on the significance of small metabolites such as sterol sulfates (StS) in the physiological regulation and mediation of programmed cell death. In this scientific process, the authors clearly isolate and identify the inhibitory compounds (consequently identified as three major sterol sulfates) extracted from differing growth stages of *S. marinoi*. They then confirmed the effect of these extracts and a synthetic standard (which also impaired the diatom growth in a dose-dependent manner) and identified a gradual increase on the intracellular concentration of these metabolites with the progression of cell growth. By analysing de novo transcriptome assemblies, the authors identified four putative enzyme groups belonging to the family sulfotransferases which control the synthesis of sterol sulphates in humans. Finally, using seven inhibitors of human sulfotransferases, only one (favonon quercetin) was observed to reduce the synthesis of sterol sulfates in *S. marinoi* cultures, resulting in the extension of growth duration and an increase in cell concentration.

The work provides strong evidence for its conclusions and has both ecological importance in terms of understanding bloom dynamics, as well as for new opportunities in the mass cultivation of microalgae. As such this manuscript provides a significant advancement in the understanding of cell regulation which I think will have uptake in both the fields of molecular ecology and algal

biotechnology. That being said, I think there should be more discussion around the application of this new knowledge, how it may link to these fields, and what are the next steps in gaining a more comprehensive understanding of programmed cell death in diatoms.

On a general note, I think this manuscript requires substantial fine tuning, especially in the area of punctuation and grammar eg. all units need to be standardised throughout (cell/mL or cells mL⁻¹); all acronyms/techniques/methods need to be full explained the first time they are mentioned; microscope magnifications are X200/x400. To assist readers outside this field I also think a schematic of the main results to accompany publication would also be of benefit. Results and Discussion need to be clearly separated, as observed for all online articles (accessed 12 May 2017).

Point-by-point response to the reviewers' comments

Text	Response
Reviewer #1 (Remarks to the Author): Diatoms and other phytoplankton are aquatic eukaryotes that generate most of the oxygen in our atmosphere and almost half the primary food chain in oceans. Despite the importance of phytoplankton to life on earth, large gaps exist in our knowledge of the diatom life cycle, which often involves massive blooms that suddenly die off and release large amounts of carbon nutrients into the ocean. After decades of studying the biochemistry of marine life, Fontana and coworkers have now discovered how sterol sulfates abruptly terminate the diatom blooms at low micromolar levels. The authors systematically searched for a hypothetical small molecule that mediates cell death in the diatom Skeletonema marinoi. Experiments involved adding methanol extracts of various cultures to healthy log-phase cultures. Addition of declining phase extracts to healthy cultures triggered symptoms of programmed cell death. Using this assay on chromatographic fractions of the declining phase extracts led to identification of sterol sulfates as substances highly associated with cell death. This finding is novel. Reports of major roles of small molecules in diatom signaling or regulation are very limited (V. Sonik et al., Mar. Drugs 13, 3672 (2015)). No sterol sulfates have been reported in diatoms apart from two vintage papers showing a non-photosynthetic diatom to contain 24-methylenecholesterol sulfate (R. Anderson et al., BBA 528, 89 (1978) and BBA 573, 557 (1979); this was the only sterol sulfate cited in the 2015 Sonik review). These preliminary results have not been extended during the past 38 years. Descriptive physiology of the rapid die-off of diatom blooms (oxidative bursts, nitric oxide production, DNA fragmentation) is well known, but the triggering of this process has been speculative, including abiotic or viral origins. The main finding was confirmed and extended: (a) Quercetin, an inhibitor of a human SULT, also inhibited the relevant SULT in S. marinoi and thus could be useful in attempts to grow cultures for biofuels or nutrients. (b) Addition of sterol sulfates to S. marinoi cultures had the same effects as normal declining phase cultures: fragmented DNA (TUNEL assays), cell aggregates (culture images), and elevated levels of nitric oxide (DAF-FM assay) and highly reactive oxygen species	

(hydroxyphenyl fluorescein assay).

The extensive experimental work is generally well done and mostly appears adequate to support the stated conclusions. However, I was confused by irregularities in making the phylogenetic tree, a basic error ("40% homology" on page 3, line 97), and the limited detail in how the de novo transcriptome was used to identify the SULT sequences. These methods should be described more accurately in greater detail. Specifically, describe (a) how sequence similarities were calculated, (b) both the full-length and pared sequence alignments used for constructing the phylogenetic tree, (c) the method for constructing the tree, (d) whether sequence identities were nucleotide or amino acid, (e) how the four SULT sequences were identified in *S. marinoi*, and (f) how the transcriptome results were analyzed. Consider calculating 1000 instead of 100 bootstraps and collapsing tree branches with low bootstrap values. Correct errors, such as two different values given for the scale bar.

The manuscript is basically well written, but the authors need to improve minor problems: definite/indefinite article usage, subject-verb plurality, and precise word choice. Getting the correct intended meaning requires author involvement. A few hours with a native English-speaking scientist should suffice to make the ca. 100 needed corrections, including misspellings, typos (e.g. page 7, line 222), and text duplications (page 9, lines 287-292).

Minor points

1. The description of SULTs on page 3, lines 87-89 should be reworded to avoid the impression that SULTs sulfonate only sterols and always use PAPS. One possibility: "Their synthesis is under control of sulfotransferases (SULT); a subfamily of these enzymes is alcohol sulfotransferases, which transfer the sulfonate moiety (-SO₃) from PAPS (3'-phosphoadenosine 5'-phosphosulfate) to the 3 β -hydroxy group of the sterol recipient."

2. The Figure 3 title is incompatible with panel a; consider moving panel a to the right side, renamed as panel d. Also, maximum likelihood is stated here but neighbor joining is mentioned in the main text (page 10, lines 306-307).

3. The sentence describing MS and NMR characterization of the isolated sterol sulfates states that "their structures were fully elucidated by 2D-NMR analysis (Supplementary Fig. S1-S4)". However, these figures show only 1D proton NMR spectra, with no

Discussion of the phylogenetic analysis has been strongly improved by implementing the description in the text and in the Methods section at pagg. 13 and 14. As requested, we have also improved the description for the identification of the SULT sequences. Additional data have been also added in the Supplementary Material about nucleotide and amino acid sequences, transcriptome analysis, SULT identification.

A new tree has been calculated by using 1000 bootstraps. The resulting graph is reported in the new Figure 3 that also contains all the other changes suggested by this reviewer.

English use has been improved and the manuscript has been read by a native English speaking colleague (Dr. Adrianna Ianora)

According to the reviewer's comments, the SULT description has been re-organized by reporting the function of these enzymes in the section "**Occurrence of sulfotransferase in marine diatoms**" and the description of the biochemical mechanism in the section "**Inhibition of sulfotransferase activity in *S. marinoi***". We hope that every vagueness has been resolved.

Figure 3 has been reorganized by separating the phylogenetic tree (revised Figure 3) from the effect due to use of SULT inhibitors (revised Figure 4).

The chemical discussion has been deeply improved by adding a new paragraph entitled "**Chemical characterization of sterol sulfates**" and by reporting NMR and MS data of the sterol sulfates in the Methods section at pag. 10 of the

numerical chemical shift information. For the benefit of researchers pursuing the growth of diatoms for biofuels and needing to chemically synthesize sterol sulfates, a typical list of NMR chemical shifts and spectral figures would be useful. Such supporting information is standard in synthetic and natural products journals, such as J Org Chem and J Nat Prod. Incidentally, the NMR figures indicate high chemical purity and high spectral quality.	revised manuscript). Furthermore, a whole set of NMR and MS spectra has been added to the Supplementary Material (Figure S1-S11) together with Table S1 that summarizes the structural assignment. Purity of natural products was determined by NMR and MS analysis, however we want to underline that most of the assays was carried out on synthetic analogs.
Reviewer #2 (Remarks to the Author): General: The manuscript describes isolation of three sterol sulfate molecules from a single marine diatom species and describes their effect on cell viability in culture. Potential genes involved in their synthesis are identified and speculation is made about their roles in broader diatom ecology. The work is novel, interesting, appears to be well done, and, once placed in proper context should prove valuable and stimulatory. There are, however, a number of significant issues. 1) The terminology and framework surrounding the processes of cell death are confused and confusing. This needs to be cleared up throughout the manuscript. For example, in the abstract, “Natural mortality” is described as a “physiological mechanism”, yet the vast majority of phytoplankton ecologists would include processes like grazing and sedimentation under ‘natural mortality’. Terms like “cell lysis”, “programmed cell death” and “apoptosis-like death” crop up and are used, almost interchangeably in places. What needs to be made clear is that: a) cell death in algae can be triggered by intrinsic as well as extrinsic factors, and b) the cell can actively participate in its own death. Whatever terminology the authors select, they should use it consistently. For one example of a consistent framework, see: Franklin, D., C. Brussaard, and J. Berges. 2006. What is the role and nature of programmed cell death in phytoplankton ecology? European Journal of Phycology 41: 1-14. 2) The background to the work is patchy. a) For example, though the work of Vardi et al. (2009) is cited, the more recently developed viral-glycosphingolipid-cell death story isn’t commented on (Vardi et al. 2012; Host-virus dynamics and subcellular controls of cell fate in a natural coccolithophore population. Proc Natl Acad Sci USA 109: 19327-19332). These authors demonstrated the induction of reactive oxygen species, caspase-specific activity, metacaspase expression, and programmed cell death in response to the	We thank this reviewer for this comment. The suggested article (European Journal of Phycology, 2006 41: 1-14) has been very useful to improve the discussion and to correct the terminology throughout the manuscript. The concept of cell death is not simple to address but we hope that we made order in the text and reduced the inconsistency of the previous version. We added to the literature the reference of Franklin, D., C. Brussaard, and J. Berges. 2006. What is the role and nature of programmed cell death in phytoplankton ecology? European Journal of Phycology 41: 1-14 We have increased the discussion about cell death in diatoms and added more references about this subject. Obviously we are aware of the work of Vardi et al. 2012 and we understand that there is a close parallelism with the results of our work. However this reference has not been included since it is not related to diatoms and we would like to keep our focus on this lineage of microalgae. Coccolithophores are very important

accumulation of virus-derived glycosphingolipids upon infection of natural E. huxleyi populations. The lipid-cell death connections seem important. b) We are told nothing about sterols in diatoms, despite the rich literature (especially from John Volkman's group, e.g. Barrett et al. 1995. Sterols of 14 species of marine diatoms. J. Phycol. 31: 360-369)...what are these compounds doing and why would they be appropriate/useful signaling compounds? c) It isn't mentioned that marine sponges (many of which graze on phytoplankton) are the richest and best-characterized sources of sterol sulphates, and the contention that steroid sulphate function "has been accurately investigated only in humans" (line 86-87) isn't quite accurate: Kobayashi et al. 1989. Hymenosulphate, a novel sterol sulphate with Ca-releasing activity from the cultured marine haptophyte Hymenomonas sp. J. Chem. Soc. Perkin Trans. 1: 101-103. 3) Culture growth phases and growth calculations. In fact, the "log" phase cultures are demonstrably not in "log" phase. Consider Figure 1. a, plotted on normal-normal axes. From day 2 to 3, cell numbers go from about 2×10^5 to 4×10^5, an approximate doubling. If the cultures were in log phase, we would then expect another doubling in the next day...to 8×10^5 by day 4. In fact, this isn't achieved until day 8. These cultures are simply not growing logarithmically by any definition. This would actually be clear if plotted on log-normal axes, a convention that is often recommended for culture data. The authors need to find a new description for the phase of growth of these cultures. In terms of growth rates expressed as a percentage of controls, it's not immediately apparent how negative numbers can be achieved (Figure 1, b and c). How were growth rates calculated? Are these in fact cell loss rates? What units are being used before they are scaled to the controls, e.g. d^{-1} using a natural or a base 10 log? This is entirely unclear. 4) The phosphatidylserine externalization experiments lack a proper control. Annexin V is a small molecule and will diffuse across 'leaky' cell membranes. Cell membranes become compromised as cell die and thus	phytoplankton components but, to the best of our knowledge, sterol sulfates have not been reported in these organisms. It will be very interesting to check if sterol sulfates have the same effect also in these protists. Sulfonation changes dramatically the chemical and biological properties of the substrates, thus sterol and sterol sulfates are biochemically distinct classes of mediators. However, in compliance with the comment of this reviewer, we have added a brief discussion about sterols and their occurrence in other organisms. A few lines have been also focused on the chemical similarities between sterols from sponges and diatoms. An deep and accurate study of the role of sterol sulfates has been reported only in humans and a few other mammals. There are other reports like the work of Kobayashi that is cited by the reviewer. However, in our opinion, this information does not allow to have a clear picture of the role of these molecules in non-mammals. We agree with the reviewer thus we have revised this part of the manuscript. The growth curve of diatoms is typically described by a logistic function. However, it is common to define log phase the ascending part of the curve. We have added this definition in the manuscript Again the reviewer is right. We have changed the axis of Figure 1b and c in order to improve the clearness of these graphs. The revised Figure 1 reports the effect of the extracts as change in cell/ml. Unfortunately the staining with propidium iodide is not possible in the diatom cells because of the interference of chlorophyll. This is a common issue in this assay but other authors have used
---	--

Annexin V become non-specific because cells stain *from the inside* without any externalization. The normal positive control is to counterstain with propidium iodide (PI). Cells staining with both PI and Annexin V are false positive, while those staining only with Annexin V can be considered to show phosphatidylserine externalization. There is no evidence that such a control was done, so the results cannot be interpreted and should not be presented. Specifics: 5) Title: “marine diatoms” is inappropriate: the authors have demonstrated an effect in a single species. They cannot generalize like this. 6) Sterol sulphates are described using trivial names; descriptions should also include proper IUPAC nomenclature...this is critical for comparisons with the literature. 7) Figure 2. b. The units of growth rate are missing from the Y axis. 8) Reference formatting needs attention. Many species names are not italicized and the titles of references are not consistently capitalized. 9) Throughout the manuscript there are grammatical issues with missing adjective articles (“a”, “the”); though annoying, these don’t generally affect interpretation.	the same protocol with Annexin V FITC for the same purpose (e.g., Iron Starvation and Culture Age Activate Metacaspases and Programmed Cell Death in the Marine Diatom Thalassiosira pseudonana. Bidle & Bender, EUKARYOTIC CELL, Feb. 2008, p. 223–236; Heat-stress-induced programmed cell death in Heterosigma akashiwo (Raphidophyceae). Jennifer E. Dingman, Janice E. Lawrence, Harmful Algae 16 (2012) 108–116). The occurrence of SULT in every diatom so far sequenced suggests that this class of enzymes can have an universal role in the lineage. If possible, we would like to underline this general aspect and keep the original title for the manuscript. IUPAC names have been added together with the chemical data in the Methods section (page 10 of the revised manuscript). Figure 2b has been corrected. References have been increased and corrected. English use has been improved and the manuscript has been read by a native English speaking colleague (Dr. Adrianna Ianora)
Reviewer #3 (Remarks to the Author): I believe this manuscript provides the first evidence on the significance of small metabolites such as sterol sulfates (StS) in the physiological regulation and mediation of programmed cell death. In this scientific process, the authors clearly isolate and identify the inhibitory compounds (consequently identified as three major sterol sulfates) extracted from differing growth stages of S. marinoi. They the confirmed the effect of these extracts and a synthetic standard (which also impaired the diatom growth in a dose-dependent manner) and identified a gradual increase on the intracellular concentration of these metabolites with	

the progression of cell growth. By analysing de novo transcriptome assemblies, the authors identified four putative enzyme groups belonging to the family sulfotransferases which control the synthesis of sterol sulphates in humans. Finally, using seven inhibitors of human sulfotransferases, only one (favonon quercetin) was observed to reduce the synthesis of sterol sulfates in *S. marinoi* cultures, resulting in the extension of growth duration and an increase in cell concentration.

The work provides strong evidence for its conclusions and has both ecological importance in terms of understanding bloom dynamics, as well as for new opportunities in the mass cultivation of microalgae. As such this manuscript provides a significant advancement in the understanding of cell regulation which I think will have uptake in both the fields of molecular ecology and algal biotechnology. That being said, I think there should be more discussion around the application of this new knowledge, how it may link to these fields, and what are the next steps in gaining a more comprehensive understanding of programmed cell death in diatoms.

On a general note, I think this manuscript requires substantial fine tuning, especially in the area of punctuation and grammar eg. all units need to be standardised throughout (cell/mL or cells mL⁻¹); all acronyms/techniques/methods need to be fully explained the first time they are mentioned; microscope magnifications are X200/x400. To assist readers outside this field I also think a schematic of the main results to accompany publication would also be of benefit. Results and Discussion need to be clearly separated, as observed for all online articles (accessed 12 May 2017).

According to this comment, we have improved the discussion about the potential use of this pathway in biotech applications with diatoms (last paragraph of the discussion, page 8 of the revised manuscript). I want to underline that the previous format of the manuscript did not allow us to address this point as it deserved.

We have checked and corrected the inconsistencies that have been highlighted by this reviewer.

As suggested by the reviewers, we added a new figure (Figure 6) in the revised manuscript to show a schematic view of our results.

Reviewers' comments:

Reviewer #1 (Remarks to the Author):

The revised manuscript is greatly improved, and the reviewer comments have largely been addressed. As may occur after such a major revision, there are many relatively minor issues and typographical errors.

Minor points:

1. Page 2, line 42: Change "remains" to "remain".
2. Page 2, line 61: Change "Fig. 1c-d" to "Fig. 1d".
3. Page 3, line 82: Change "In the high resolution electrospray mass analysis" to "by high resolution electrospray ionization mass spectrometry" to match the abbreviation HR-ESI-MS.
4. Page 3, line 84: Change the first "0.84" to "0.83" to match values in Table S1.
5. Page 3, line 87: Change "methylene group" to "C4 protons". See Wikipedia "methylene group" for ambiguity and conflict about use of "methylene group".
6. Page 3, line 93: The references "(Uomori et al., 1992; Nes et al., 1975)", apart from not being cited correctly, are not found among the references. (a) The 1975 references of William D. Nes are not highly appropriate here and use outdated NMR methods. The only relevant 1992 Uomori reference (Uomori, A.; Nakagawa, Y.; Yoshimatsu, S.; Sep, S.; Sankawa, U.; Takeda, K. *Phytochemistry* 1992, 31, 1569-1572) has in Figure 3 an excellent comparison of the assigned ¹H NMR methyl signals of campesterol and dihydrobrassicasterol in CDCl₃. Even neglecting signal assignments, the figure shows a pair of closely spaced doublets at ca. 0.78 ppm for dihydrobrassicasterol and more broadly spaced doublets for campesterol. The pair of closely spaced doublets of the sulfate derivatives in CD₃OD solution (Fig. S4 and Table S1) are thus strongly suggestive of dihydrobrassicasterol sulfate. (b) However, the NMR spectra of the 3β-sulfates here were acquired in CD₃OD solution, whereas literature spectra of 3β-hydroxysterols were measured in CDCl₃ solution. The big change in solvent polarity results in substantial and uneven changes in chemical shift, depending on the differences in solvation of the methyl groups. Although the authors probably prepared the sulfate of dihydrobrassicasterol from a sample of the free sterol and thus had an NMR spectrum of known dihydrobrassicasterol sulfate, I did not want to raise this issue and perhaps create another round of revisions. So I obtained ¹H, ¹³C, DEPT, and HSQC spectra of commercial sitosterol (55% sitosterol, 30% campesterol, and 15% dihydrobrassicasterol) in CDCl₃ and CD₃OD. Reliable ¹H and ¹³C assignments were obtained for all methyl signals in both solvents. As expected, chemical shifts were quite different in the two solvents. However, the close spacing of the pair of doublets in CDCl₃ was maintained fairly well in CD₃OD. This result validated the authors' conclusion that the major sterol sulfate in *S. marinoi* is brassicasterol sulfate. (c) To expedite this revision, it is suggested to ignore my minor concern by not modifying any wording in lines 88-92. For references on line 93, use the above Uomori citation and "Goad, L. J.; Akihisa, T. *Analysis of Sterols*; Blackie (Chapman & Hall): London, 1997, p 367-368.", which contains an excellent compendium of high-precision ¹H and ¹³C NMR chemical shifts for campesterol and dihydrobrassicasterol in CDCl₃. (d) Given the many reference additions and renumbering, the authors should carefully check the accuracy of all reference numbering.
7. Page 3, line 94: Delete "spectrum" (a redundant word after "MS"). Also, consider using hyphens in the abbreviation HR-ESI-MS.
8. Page 4, line 107: Clarify the nature and origin of the StS mixture. Was this a natural mixture isolated from *S. marinoi* or a synthetic mixture? In the latter case, give the ratio of StS components.
9. Page 4, line 124: The term "acceptor" is ambiguous (an electron acceptor or an enzyme or a steroid substrate?). This paragraph should be rewritten as an introduction for non-experts. Start by describing the broad range of SULT functions, their intracellular locations, and their diversity of acceptor molecules. Then focus on the cytosolic subset of SULTs in humans that sulfonate steroids and other small molecules. With this background, the reader can understand the term "cytosolic sulfotransferase superfamily" and the phylogenetic tree in Fig. 3.
10. Page 4, line 130: Delete the comma after "several".
11. Pages 7-8 (Discussion): Beyond the first 15 lines, 90% of the Discussion comprises new

sentences of a somewhat speculative nature, well beyond the experimental results. Such additions can be welcome if they are concise and focused on insights derived from the new findings.

However, much of this later Discussion comprises clumsy thinking without a clear goal, e.g. page 8, lines 251-252 "the growth rates of microalgae may also contribute to mitigate carbon dioxide". Basically, the Discussion needs marked improvement and/or marked shortening.

12. Page 8, lines 252-254: This sentence needs a reference.

13. Page 10, lines 316-317, 322-324, 329-331: The IUPAC names all have errors, e.g. lacking the "8S,9S" of the tetracycle, including a superfluous "10" and "13" (quaternary positions) for the "dodecahydro" numbering (given incorrectly as "tetradecahydro"), and attaching the C1 instead of C2 of the side chain to the tetracycle. IUPAC names could be copied online for cholesterol sulfate (Wikipedia and PubChem) and sitosterol sulfate (PubChem only). For dihydrobrassicasterol, the sulfate IUPAC name is not so easily found, but can be obtained by modifying the PubChem name for dihydrobrassicasterol by adding initial and terminal brackets, changing "3-ol" to "3-yl", and adding "hydrogen sulfate". However, I don't understand why the dimethyl substituent precedes the side chain substituent for cholesterol sulfate but not for the other sulfates. The (hopefully) correct IUPAC names are:

[(3S,8S,9S,10R,13R,14S,17R)-10,13-Dimethyl-17-[(2R)-6-methylhepta n-2-yl]-2,3,4,7,8,9,11,12,14,15,16,17-dodecahydro-1H-cyclopenta[a]phenanthren-3-yl] hydrogen sulfate (cholesterol sulfate);

[(3S,8S,9S,10R,13R,14S,17R)-17-[(2R,5R)-5-ethyl-6-methylheptan-2-yl]-10,13-dimethyl-2,3,4,7,8,9,11,12,14,15,16,17-dodecahydro-1H-cyclopenta[a]phenanthren-3-yl] hydrogen sulfate (sitosterol sulfate)

[(3S,8S,9S,10R,13R,14S,17R)-17-[(2R,5S)-5,6-dimethylheptan-2-yl]-10,13-dimethyl-2,3,4,7,8,9,11,12,14,15,16,17-dodecahydro-1H-cyclopenta[a]phenanthren-3-yl] hydrogen sulfate (dihydrobrassicasterol sulfate)

Given the limited value of these IUPAC names (largely available online by searching the common name) plus the significant chance of errors during reviewer input, author response, and typesetting, it might be preferable to skip the IUPAC names.

14. Page 10, lines 315-335. The 1H NMR data presented here duplicate data in Table S1 and should be deleted. The ESI-MS data should similarly be moved to Figures S1-S3. Provide a note to see the 1H NMR and ESI-MS data in Supplementary Information.

15. Pages 12-14: The improvements to the sections "RNA sequencing" and "SULT gene analysis" are greatly appreciated.

16. Page 23, Figure 1: (a) Do not use a continuous X-axis in panels b and c to avoid the impression that none of the bars are exactly at 20, 40, or 60. (b) Line 736: Change "24h and 48h" to "24 h (b)" and "48 h (c)". (c) Line 737: Add "(20, 40, and 60 ug/mL)" after "MeOH". (d) Line 738: Change "only treated" to "treated only".

17. Page 24, Figure 2: (a) In the table of panel a, delete the column headed "R" or replace "R" with C-24 substituent. (b) Lines 750-751: Are these natural mixtures synthetic or from yeast extracts? (c) Line 756: I don't understand how the "three different cell densities" are indicated in panel b and what the horizontal lines mean in panel b. (d) Line 758: I don't understand which are the "Four independent experiments" and the three biological replicates.

18. Page 25, lines 768-770: This sentence suggests that Maximum Likelihood method was used to align the 172 amino acids, whereas the section on "SULT gene analysis" indicates that alignment was done by ClustalW2 software (page 13, line 435). This introductory sentence should be succinct and accurate. The entire Figure legend could be more condensed and focused.

19. Please check for further errors, as I had time to check only half the manuscript.

Reviewer #2 (Remarks to the Author):

The revised manuscript describes isolation of three sterol sulfate molecules from a single marine diatom species and their effect on cell viability in culture. Potential genes involved in their synthesis are identified and speculation is made about their roles in broader diatom ecology. The

work is novel, interesting, appears to be well done. The revision has addressed a number of issues and is significantly improved. A couple points remain that are still problematic.

1. Original point: The terminology and framework surrounding the processes of cell death are confused and confusing. Authors' Reply: We thank this reviewer for this comment. The suggested article (European Journal of Phycology, 2006 41: 1-14) has been very useful to improve the discussion and to correct the terminology throughout the manuscript. The concept of cell death is not simple to address but we hope that we made order in the text and reduced the inconsistency of the previous version.

The revision is clearer and the terminology, though not entirely to my taste is consistent.

2. Original point: The background to the work is patchy. For example, though the work of Vardi et al. (2009) is cited, the more recently developed viral-glycosphingolipid cell death story isn't commented on (Vardi et al. 2012; Host-virus dynamics and subcellular controls of cell fate in a natural coccolithophore population. Proc Natl Acad Sci USA 109: 19327-19332). Author's Reply: We have increased the discussion about cell death in diatoms and added more references about this subject. Obviously we are aware of the work of Vardi et al. 2012 and we understand that there is a close parallelism with the results of our work. However this reference has not been included since it is not related to diatoms and we would like to keep our focus on this lineage of microalgae.

It seems like a missed opportunity, but that's the authors' choice.

3. Original point: It isn't mentioned that marine sponges (many of which graze on phytoplankton) are the richest and best characterized sources of sterol sulphates, and the contention that steroid sulphate function "has been accurately investigated only in humans" (line 86-87) isn't quite accurate: Kobayashi et al. 1989. Hymenosulphate, a novel sterol sulphate with Ca releasing activity from the cultured marine haptophyte *Hymenomonas* sp. J. Chem. Soc. Perkin Trans. 1: 101-103. Authors' reply: A few lines have been also focused on the chemical similarities between sterols from sponges and diatoms. A deep and accurate study of the role of sterol sulfates has been reported only in humans and a few other mammals. There are other reports like the work of Kobayashi that is cited by the reviewer. However, in our opinion, this information does not allow to have a clear picture of the role of these molecules in nonmammals.

The additional sterol discussion is well done. On the other hand, it seems a terrible idea to ignore other reports of sterol sulphate in algae, especially ones like Kobayashi that describe functional aspects. Calcium release is after all associated with cell death! Such work deserves to be cited, even if it detracts from the novelty of the present manuscript.

4. Original point: Culture growth phases and growth calculations. In fact, the "log" phase cultures are demonstrably not in "log" phase. Author's reply: We agree with the reviewer thus we have revised this part of the manuscript. The growth curve of diatoms is typically described by a logistic function. However, it is common to define log phase the ascending part of the curve. We have added this definition in the manuscript.

The authors have missed the point here. Whatever function is used to describe the curve, the cells are demonstrably NOT in log phase at the point of sampling. Again, look at Figure 1 a. If the cells depicted in "LPEx" on day 6 were actually in log phase, we would expect them to be able to divide at least one more time, i.e. from 8×10^5 to 16×10^5 cell/ml. They do not. They are NOT in log phase. QED. Certainly, many people get this wrong, but the authors should know better. Now, how important is this for the manuscript? Maybe not a lot...we see a difference between what we could call "early senescent phase" (how about "eSPEx"?) and "late senescent phase" (how about "ISPEx"?) and the declining phase. But please, let's not pretend that this is log phase, because it is not.

5. Original Point: The phosphatidylserine externalization experiments lack a proper control. Annexin V is a small molecule and will diffuse across 'leaky' cell membranes. Authors' reply: Unfortunately the staining with propidium iodide is not possible in the diatom cells because of the interference of chlorophyll. This is a common issue in this assay but other authors have used Annexin V...

The point remains unaddressed. Without a control, the Annexin staining tells us nothing. This is well understood by those who do Annexin staining. It might be that the cells have inverted PS on the membrane or it might be that the membrane is permeable. The authors are quite correct that a PI control is problematic in chlorophyll-containing cells. Other authors have chosen to run parallel controls, staining for membrane permeability and subtracting proportions of cells with compromised membranes from those staining with Annexin to derive an index of true Annexin staining. But as it stands, the results cannot be presented as evidence of a cell death process. That others have made the same error isn't a rationale for repeating it.

6. Original point: Title: "marine diatoms" is inappropriate: the authors have demonstrated an effect in a single species. They cannot generalize like this. Authors' reply: The occurrence of SULT in every diatom so far sequenced suggests that this class of enzymes can have an universal role in the lineage. If possible, we would like to underline this general aspect and keep the original title for the manuscript.

If the title merely talked about the presence of the compounds, that would be fine...but it argues that the sterols sulfates mediate a cell death programme. This has only been shown in one diatom....and it is speculation that it a conserved phenomenon. Certainly, speculate in the discussion, but the title needs to be factual..."diatoms" is unsupportable.

Reviewer #3 (Remarks to the Author):

I believe your point by point responses to reviewers comments have been adequately addressed and would agree that this manuscript be published in its revised form.

	Reviewer #1 (Remarks to the Author)	RESPONSE
1-5	1. Page 2, line 42: Change “remains” to “remain”. 2. Page 2, line 61: Change “Fig. 1c-d” to “Fig. 1d”. 3. Page 3, line 82: Change “In the high resolution electrospray mass analysis” to “by high resolution electrospray ionization mass spectrometry” to match the abbreviation HR-ESI-MS. 4. Page 3, line 84: Change the first “0.84” to “0.83” to match values in Table S1. 5. Page3, line 87: Change “methylene group” to “C4 protons”. See Wikipedia “methylene group” for ambiguity and conflict about use of “methylene group”.	Done
6	6. Page 3, line 93: The references “(Uomori et al., 1992; Nes et al., 1975)”, apart from not being cited correctly, are not found among the references. (a) The 1975 references of William D. Nes are not highly appropriate here and use outdated NMR methods. The only relevant 1992 Uomori reference (Uomori, A.; Nakagawa, Y.; Yoshimatsu, S.; Sep, S.; Sankawa, U.; Takeda, K. Phytochemistry 1992, 31, 1569-1572) has in Figure 3 an excellent comparison of the assigned ¹H NMR methyl signals of campesterol and dihydrobrassicasterol in CDCl₃. Even neglecting signal assignments, the figure shows a pair of closely spaced doublets at ca. 0.78 ppm for dihydrobrassicasterol and more broadly spaced doublets for campesterol. The pair of closely spaced doublets of the sulfate derivatives in CD₃OD solution (Fig. S4 and Table S1) are thus strongly suggestive of dihydrobrassicasterol sulfate. (b) However, the NMR spectra of the 3β-sulfates here were acquired in CD₃OD solution, whereas literature spectra of 3β-hydroxysterols were measured in CDCl₃ solution. The big change in solvent polarity results in substantial and uneven changes in chemical shift, depending on the differences in solvation of the methyl groups. Although the authors probably prepared the sulfate of dihydrobrassicasterol from a sample of the free sterol and thus had an NMR spectrum of known dihydrobrassicasterol sulfate, I did not want to raise this issue and perhaps create another round of revisions. So I obtained ¹H, ¹³C, DEPT, and HSQC spectra of commercial sitosterol (55% sitosterol, 30% campesterol, and 15% dihydrobrassicasterol) in CDCl₃ and CD₃OD. Reliable ¹H and ¹³C assignments were obtained for all methyl signals in both solvents. As expected, chemical shifts were quite different in the two solvents. However, the close spacing of the pair of doublets in CDCl₃ was maintained fairly well in CD₃OD. This result validated the authors’ conclusion that the major sterol sulfate in S. marinoi is brassicasterol sulfate. (c) To expedite this revision, it is suggested to ignore my minor concern by not modifying any wording in lines	We have greatly appreciated the careful analysis made by this reviewer. This is not usual and we want to thank her/him for the efforts and time spent on this review. Changes have been made and the list of references has been updated and checked.

	88-92. For references on line 93, use the above Uomori citation and "Goad, L. J.; Akihisa, T. Analysis of Sterols; Blackie (Chapman & Hall): London, 1997, p 367-368.", which contains an excellent compendium of high-precision ¹ H and ¹³ C NMR chemical shifts for campesterol and dihydrobrassicasterol in CDCl ₃ . (d) Given the many reference additions and renumbering, the authors should carefully check the accuracy of all reference numbering.	
7	7. Page 3, line 94: Delete "spectrum" (a redundant word after "MS"). Also, consider using hyphens in the abbreviation HR-ESI-MS.	Done
8	8. Page 4, line 107: Clarify the nature and origin of the StS mixture. Was this a natural mixture isolated from S. marinoi or a synthetic mixture? In the latter case, give the ratio of StS components.	We referred to the natural products. The point has been clarified.
9	9. Page 4, line 124: The term "acceptor" is ambiguous (an electron acceptor or an enzyme or a steroid substrate?). This paragraph should be rewritten as an introduction for non-experts. Start by describing the broad range of SULT functions, their intracellular locations, and their diversity of acceptor molecules. Then focus on the cytosolic subset of SULTs in humans that sulfonate steroids and other small molecules. With this background, the reader can understand the term "cytosolic sulfotransferase superfamily" and the phylogenetic tree in Fig. 3.	This paragraph has been rewritten according to the reviewer's suggestions.
10	10. Page 4, line 130: Delete the comma after "several".	Done
11	11. Pages 7-8 (Discussion): Beyond the first 15 lines, 90% of the Discussion comprises new sentences of a somewhat speculative nature, well beyond the experimental results. Such additions can be welcome if they are concise and focused on insights derived from the new findings. However, much of this later Discussion comprises clumsy thinking without a clear goal, e.g. page 8, lines 251-252 "the growth rates of microalgae may also contribute to mitigate carbon dioxide". Basically, the Discussion needs marked improvement and/or marked shortening.	I have shortened the discussion as much as I could. However, most of these comments were introduced in response to explicit requests of the other reviewers. A new reference has been introduced about algae and carbon mitigation
12	12. Page 8, lines 252-254: This sentence needs a reference.	done
13	13. Page 10, lines 316-317, 322-324, 329-331: The IUPAC names all have errors, e.g. lacking the "8S,9S" of the tetracycle, including a superfluous "10" and "13" (quaternary positions) for the "dodecahydro" numbering (given incorrectly as "tetradecahydro"), and attaching the C1 instead of C2 of the side chain to the tetracycle. IUPAC names could be copied online for cholesterol sulfate (Wikipedia and PubChem) and sitosterol sulfate (PubChem	IUPAC name have been deleted.

	only). For dihydrobrassicasterol, the sulfate IUPAC name is not so easily found, but can be obtained by modifying the PubChem name for dihydrobrassicasterol by adding initial and terminal brackets, changing “3-ol” to “3-yl”, and adding “hydrogen sulfate”. However, I don’t understand why the dimethyl substituent precedes the side chain substituent for cholesterol sulfate but not for the other sulfates. The (hopefully) correct IUPAC names are: [[3S,8S,9S,10R,13R,14S,17R)-10,13-Dimethyl-17-[(2R)-6-methylheptan-2-yl]-2,3,4,7,8,9,11,12,14,15,16,17-dodecahydro-1H-cyclopenta[a]phenanthren-3-yl] hydrogen sulfate (cholesterol sulfate); [[3S,8S,9S,10R,13R,14S,17R)-17-[(2R,5R)-5-ethyl-6-methylheptan-2-yl]-10,13-dimethyl-2,3,4,7,8,9,11,12,14,15,16,17-dodecahydro-1H-cyclopenta[a]phenanthren-3-yl] hydrogen sulfate (sitosterol sulfate) [[3S,8S,9S,10R,13R,14S,17R)-17-[(2R,5S)-5,6-dimethylheptan-2-yl]-10,13-dimethyl-2,3,4,7,8,9,11,12,14,15,16,17-dodecahydro-1H-cyclopenta[a]phenanthren-3-yl] hydrogen sulfate (dihydrobrassicasterol sulfate) Given the limited value of these IUPAC names (largely available online by searching the common name) plus the significant chance of errors during reviewer input, author response, and typesetting, it might be preferable to skip the IUPAC names.	
14	14. Page 10, lines 315-335. The 1H NMR data presented here duplicate data in Table S1 and should be deleted. The ESI-MS data should similarly be moved to Figures S1-S3. Provide a note to see the 1H NMR and ESI-MS data in Supplementary Information.	The presentation of the spectroscopic data has been revised according to the reviewer's comment.
15	15. Pages 12-14: The improvements to the sections “RNA sequencing” and “SULT gene analysis” are greatly appreciated.	Thanks
16	16. Page 23, Figure 1: (a) Do not use a continuous X-axis in panels b and c to avoid the impression that none of the bars are exactly at 20, 40, or 60. (b) Line 736: Change “24h and 48h” to “24 h (b)” and “48 h (c)”. (c) Line 737: Add “(20, 40, and 60 ug/mL)” after “MeOH”. (d) Line 738: Change “only treated” to “treated only”.	Done
17	17. Page 24, Figure 2: (a) In the table of panel a, delete the column headed “R” or replace “R” with C-24 substituent. (b) Lines 750-751: Are these natural mixtures synthetic or from yeast extracts? (c) Line 756: I don’t understand how the “three different cell densities” are indicated in panel b and what the horizontal lines mean in panel b. (d) Line 758: I don’t understand which are the “Four independent experiments” and the three biological replicates.	Text has been revised and the clearness has been improved

18	18. Page 25, lines 768-770: This sentence suggests that Maximum Likelihood method was used to align the 172 amino acids, whereas the section on "SULT gene analysis" indicates that alignment was done by ClustalW2 software (page 13, line 435). This introductory sentence should be succinct and accurate. The entire Figure legend could be more condensed and focused.	The mistake has been corrected. and the figure legend has been shortened.
19	19. Please check for further errors, as I had time to check only half the manuscript.	Done

	Reviewer #2 (Remarks to the Author)	RESPONSE
1	The revised manuscript describes isolation of three sterol sulfate molecules from a single marine diatom species and their effect on cell viability in culture. Potential genes involved in their synthesis are identified and speculation is made about their roles in broader diatom ecology. The work is novel, interesting, appears to be well done. The revision has addressed a number of issues and is significantly improved. A couple points remain that are still problematic. 1. Original point: The terminology and framework surrounding the processes of cell death are confused and confusing. Authors' Reply: We thank this reviewer for this comment. The suggested article (European Journal of Phycology, 2006 41: 1-14) has been very useful to improve the discussion and to correct the terminology throughout the manuscript. The concept of cell death is not simple to address but we hope that we made order in the text and reduced the inconsistency of the previous version. The revision is clearer and the terminology, though not entirely to my taste is consistent.	OK
2	2. Original point: The background to the work is patchy. For example, though the work of Vardi et al. (2009) is cited, the more recently developed viral-glycosphingolipid cell death story isn't commented on (Vardi et al. 2012; Host-virus dynamics and subcellular controls of cell fate in a natural coccolithophore population. Proc Natl Acad Sci USA 109: 19327-19332). Author's Reply: We have increased the discussion about cell death in diatoms and added more references about this subject. Obviously we are aware of the work of Vardi et al. 2012 and we understand that there is a close parallelism with the results of our work. However this reference has not been included since it is not related to diatoms and we would like to keep our focus on this lineage of microalgae. It seems like a missed opportunity, but that's the authors' choice.	We have added the reference to Kobayashi et al. on the presence of sterol sulfate in coccolithophores (see point 3 below) and we take the opportunity to cite the work of Vardi on the effect of sphingolipids. Thus, the paper " Vardi et al. 2012" has been added to the revised manuscript.
3	3. Original point: It isn't mentioned that marine sponges (many of which graze on phytoplankton) are the richest and best characterized sources of sterol sulphates, and the contention that steroid sulphate function "has been accurately investigated only in	As require by the reviewer we have slightly improved this part. In particular, we have introduced a short

	humans" (line 86-87) isn't quite accurate: Kobayashi et al. 1989. Hymenosulphate, a novel sterol sulphate with Ca releasing activity from the cultured marine haptophyte Hymenomonas sp. J. Chem. Soc. Perkin Trans. 1: 101-103. Authors' reply: A few lines have been also focused on the chemical similarities between sterols from sponges and diatoms. An deep and accurate study of the role of sterol sulfates has been reported only in humans and a few other mammalians. There are other reports like the work of Kobayashi that is cited by the reviewer. However, in our opinion, this information does not allow to have a clear picture of the role of these molecules in nonmammalians. The additional sterol discussion is well done. On the other hand, it seems a terrible idea to ignore other reports of sterol sulphate in algae, especially ones like Kobayashi that describe functional aspects. Calcium release is after all associated with cell death! Such work deserves to be cited, even if it detracts from the novelty of the present manuscript.	discussion about the work of Kobayashi on Hymenosulphate and added this paper to the references of the revised manuscript.
4	4. Original point: Culture growth phases and growth calculations. In fact, the "log" phase cultures are demonstrably not in "log" phase. Author's reply: We agree with the reviewer thus we have revised this part of the manuscript. The growth curve of diatoms is typically described by a logistic function. However, it is common to define log phase the ascending part of the curve. We have added this definition in the manuscript. The authors have missed the point here. Whatever function is used to describe the curve, the cells are demonstrably NOT in log phase at the point of sampling. Again, look at Figure 1 a. If the cells depicted in "LPEX" on day 6 were actually in log phase, we would expect them to be able to divide at least one more time, i.e. from 8×10^5 to 16×10^5 cell/ml. They do not. They are NOT in log phase. QED. Certainly, many people get this wrong, but the authors should know better. Now, how important is this for the manuscript? Maybe not a lot...we see a difference between what we could call "early senescent phase" (how about "eSPEX"?) and "late senescent phase" (how about "ISPEX"?) and the declining phase. But please, let's not pretend that this is log phase, because it is not.	Thanks for the explanation. We have revised the text and we named the point as "End of the Log Phase" that should clarify that we were not in log phase when we collected the cells.
5	5. Original Point: The phosphatidylserine externalization	Many thanks. We respect

	experiments lack a proper control. Annexin V is a small molecule and will diffuse across 'leaky' cell membranes. Authors' reply: Unfortunately the staining with propidium iodide is not possible in the diatom cells because of the interference of chlorophyll. This is a common issue in this assay but other authors have used Annexin V... The point remains unaddressed. Without a control, the Annexin staining tells us nothing. This is well understood by those who do Annexin staining. It might be that the cells have inverted PS on the membrane or it might be that the membrane is permeable. The authors are quite correct that a PI control is problematic in chlorophyll-containing cells. Other authors have chosen to run parallel controls, staining for membrane permeability and subtracting proportions of cells with compromised membranes from those staining with Annexin to derive an index of true Annexin staining. But as it stands, the results cannot be presented as evidence of a cell death process. That others have made the same error isn't a rationale for repeating it.	and agree with the reviewer's comments. It was not our intention to force the meaning of our results. The text has been changed and the Annexin staining is shown as a supporting approach with the warning that this result cannot be considered conclusive and needs further demonstration.
6	6. Original point: Title: "marine diatoms" is inappropriate: the authors have demonstrated an effect in a single species. They cannot generalize like this. Authors' reply: The occurrence of SULT in every diatom so far sequenced suggests that this class of enzymes can have an universal role in the lineage. If possible, we would like to underline this general aspect and keep the original title for the manuscript. If the title merely talked about the presence of the compounds, that would be fine...but it argues that the sterols sulfates mediate a cell death programme. This has only been shown in one diatom....and it is speculation that it a conserved phenomenon. Certainly, speculate in the discussion, but the title needs to be factual..."diatoms" is unsupportable.	We changed the title and the work is now referred to a single bloom-forming diatom species.